# Comparative characterization of the infant gut microbiome and their maternal lineage by a multi-omics approach

Tomás Clive Barker-Tejeda [1,2,12], Elisa Zubeldia-Varela [1,2,12], Andrea Macías-Camero [1,2,12], Lola Alonso [3], Isabel Adoración Martín-Antoniano[2,3,4], María Fernanda Rey-Stolle [1], Leticia Mera-Berriatua[2], Raphaëlle Bazire[5,6], Paula Cabrera-Freitag [7], Meera Shanmuganathan [8], Philip Britz-McKibbin [8], Carles Ubeda [9,10], M. Pilar Francino [10,11], Domingo Barber [2], María Dolores Ibáñez-Sandín[5,6], Coral Barbas [1], Marina Pérez-Gordo [2] ✉ & Alma Villaseñor [1,2] ✉

The human gut microbiome establishes and matures during infancy, and dysregulation at this stage may lead to pathologies later in life. We conducted a multi-omics study comprising three generations of family members to investigate the early development of the gut microbiota. Fecal samples from 200 individuals, including infants (0-12 months old; 55% females, 45% males) and their respective mothers and grandmothers, were analyzed using two independent metabolomics platforms and metagenomics. For metabolomics, gas chromatography and capillary electrophoresis coupled to mass spectrometry were applied. For metagenomics, both 16S rRNA gene and shotgun sequencing were performed. Here we show that infants greatly vary from their elders in fecal microbiota populations, function, and metabolome. Infants have a less diverse microbiota than adults and present differences in several metabolite classes, such as short- and branched-chain fatty acids, which are associated with shifts in bacterial populations. These findings provide innovative biochemical insights into the shaping of the gut microbiome within the same generational line that could be beneficial in improving childhood health outcomes.

The human gut microbiota is highly dynamic and evolves throughout the lifespan, mostly in the first three years of life. From birth, there is a symbiotic crosstalk between the microbiota and human cells, adapting to changes over time. The gut microbiota is variable among individuals because it is highly dependent on host-associated factors such as diet, lifestyle, environmental factors, and age, among others[1–3]. Because of its significant between-subject variability, it is difficult to define what constitutes a "normal" or "beneficial" microbiota, although it is generally considered healthier the greater its diversity and balance between species[4]. In addition, its dysregulation – i.e., dysbiosis – at early stages of life can be implicated in the development of pathologies later in life[5–9].

The initial microbial seeding of the infant comes predominantly from the maternal side since vaginal delivery and breastfeeding influence the primary colonization of the gut microbiota, among other factors[10–14]. In addition, it has also been discussed that pre-birth colonization could exist during pregnancy in utero; however, this remains unclear[15]. Furthermore, it has been shown that the first 1000 days of life, when newborns first encounter environmental exposures, are critical for the development of the intestinal microbiota[16]. This decisive timeframe is an underexplored area with great potential for personalized medicine interventions. Despite the emerging data, the infant gut microbiota and their metabolic products are still poorly characterized,

and little is still known about its changes with age and how this process is influenced by the maternal line.

Omics methodologies have become an essential tool in the characterization of the human microbiome, including high-throughput technologies which allow the screening of large amounts of biological data at different levels from different organisms. Indeed, genomics has been used widely to investigate the composition of microbial ecosystems – e.g., with 16S ribosomal RNA (rRNA) gene sequencing[17] – and their functions – e.g., with shotgun metagenomics sequencing[18,19]. Several studies have applied genomics to analyze the microbiome in infants and their mothers and, to a lesser extent, the elderly generations[20–22]. However, this only provides information about the bacterial taxonomy, and few studies have complemented genomics with metabolomics, which is crucial in understanding the complex metabolic interactions between gut microbiota and the host as modulated by environmental exposures[23,24]. Furthermore, from those related to aging, none of them have included three generations and more than one platform for each omic as this study does[25–30].

A valuable metabolomics technique in this regard is gas chromatography coupled to mass spectrometry (GC-MS), as it can reliably measure a wide range of metabolites associated with host-gut microbiota co-metabolism[31–33]. However, a common drawback in the case of non-volatile biological samples is that the sample preparation is usually time-consuming compared to other separation techniques, especially in large-scale studies[32,34,35]. Consequently, alternative methods have appeared, such as alkyl chloroformate derivatization[36], which is faster and can be performed at room temperature in aqueous media[31,37]. These noticeable advantages make it desirable for studies with larger numbers of samples.

Another technique capable of detecting metabolites of interest –such as short-chain fatty acids (SCFAs), amino acids, and other charged metabolites– is capillary electrophoresis coupled to mass spectrometry (CE-MS)[33]. An innovative application of CE-MS especially designed for large cohorts is multisegment injection (MSI-CE-MS)[38], which allows the measurement of up to 13 samples in a single run. This results in a higher throughput for large-scale analyses (≈4 min/sample) and has recently shown remarkable advantages in metabolomics studies[38–41]. Taking all this into account, we propose here a methodology for large-scale GC-MS analyses using a quadrupole-time-of-flight (QTOF) mass analyzer (GC-QTOF-MS) and complement these results with a targeted search in MSI-CE-TOF-MS.

To date, no study has employed a multi-omics approach to characterize in detail the gut microbiome, its functions, and its relationship to the fecal metabolome in young infants compared to their maternal lineage including two elder generations. In this work, we report a large-scale, multi-omics analysis of a three-generation cohort of 200 fecal samples. We recruited infants younger than twelve months of age, their mothers, and maternal grandmothers, and applied metabolomics (untargeted GC-QTOF-MS and targeted MSI-CE-MS) and metagenomics (16S rRNA gene and shotgun sequencing). Furthermore, we integrated the results from these techniques using multivariant analysis with the mixOmics package.

Here, we show the differences in the gut metabolome and metagenomes of 3 consecutive family generations and demonstrate the unique multi-omics signature of young infants. Our study provides interesting insights into the development of the microbiome and its related metabolites throughout the lifespan that could serve to improve childhood health outcomes.

## Results

### Study population and final sample numbers for each analysis
The study population characteristics are shown in Table S1. The mean age (years ± SD) of the groups was: Infants (0.42 ± 0.14; $n = 69$), Mothers (34.31 ± 3.8; $n = 67$) and Grandmothers (63.09 ± 6.09; $n = 64$). In each step of the analysis, due to insufficient quantity, equipment malfunction, or outlier removal, a different final number of samples was obtained. The justification for these differences in sample numbers is presented in detail in Fig. S1A for each analysis and in Fig. S1B for the multi-omics integration. More specifically, those participants with more extreme ages in each age group were deemed age outliers and excluded from further analysis (Fig. S2).

### Large-scale sample analysis using GC-QTOF-MS and successful batch strategy
To measure our cohort of samples in a large-scale GC-MS analysis, we adapted a short derivatization method (<15 min) for sample treatment[31,42]. To complement this methodology, samples were analyzed within a batch strategy using a common quality control sample (QC) made from pooled samples from the three age groups and analyzed during all batches (Fig. 1A). We split the analysis in three batches according to the age group (Infants, Mothers, and Grandmothers), and the QC sample was used to normalize the data to correct analytical drift and batch effects.

Data deconvolution and metabolite annotation – as detailed in Methods – resulted in 152 molecular features obtained in all samples. After quality assurance (QA), 134 features remained among the three batches, as shown in Fig. S3A. Intra-batch and inter-batch normalizations[43–45] were applied after obtaining the principal component analysis (PCA) model of the raw data (Fig. 1B). This normalization strategy resulted in the clustering of the QCs from the three batches (Fig. 1C). Other quality parameters that improved with the normalization were the total useful signal (TUS) and the relative standard deviation (RSD) of the two internal standards (IS): 4-methyl valeric (4MV, IS1) for controlling the derivatization process and tricosane (TRI, IS2) for instrumental performance (Fig. S3B, D). Additionally, a higher number of features cleared the QA after normalization: 146 vs 134 passed in at least one batch, and 110 vs 65 in all three batches –almost 1.7 times more features– (Fig. S3C).

To sum up, the analytical method and the QC strategy coupled to the normalization allowed the successful joining of the three batches, resulting in a decrease of the sample outliers compared to before normalization (Fig. 1C). In addition, Infant samples clustered apart from the adults in the PCA analysis, suggesting a great difference between the metabolomes of Infants and adults.

### Analysis of selected fecal metabolites by MSI-CE-MS
Complementarily to the GC-QTOF-MS analysis, we obtained a total of 14 key metabolites of the host-gut microbiota co-metabolism with the MSI-CE-MS methodology after a targeted search (Fig. S4A), as detailed in Methods. Of these, 6 were in common with GC-QTOF-MS to check the accordance between both techniques (Fig. S5), while the other 8 were crucial metabolites that had not been previously detected. The justification of this selection is shown in Fig. S4B, while the PCA that shows the successful clustering of the QCs is shown in Fig. S4C.

### Fecal metabolome differences between the age groups
After checking the quality of the data and the success of both metabolomics platforms, we focused further on determining the differences between the three age groups.

To address the dependence between samples (Infants, Mothers, and Grandmothers), and the situation of relatives, we applied univariate statistics using a linear mixed-effects model (GLM) to both datasets. Regarding the GC-QTOF-MS data, a total of 77 metabolites showed statistical variation among age groups (FDR < 0.05, Supplementary Data 1). As expected, higher numbers of differences were observed between Infants and their Mothers ($n = 67$), and Infants and their Grandmothers ($n = 62$) than between Mothers and Grandmothers ($n = 29$). These variations were compared in a Venn diagram (Fig. S6A) and are presented in Supplementary Data 1 with their biochemical information. Among the significant metabolite classes, amino acids,

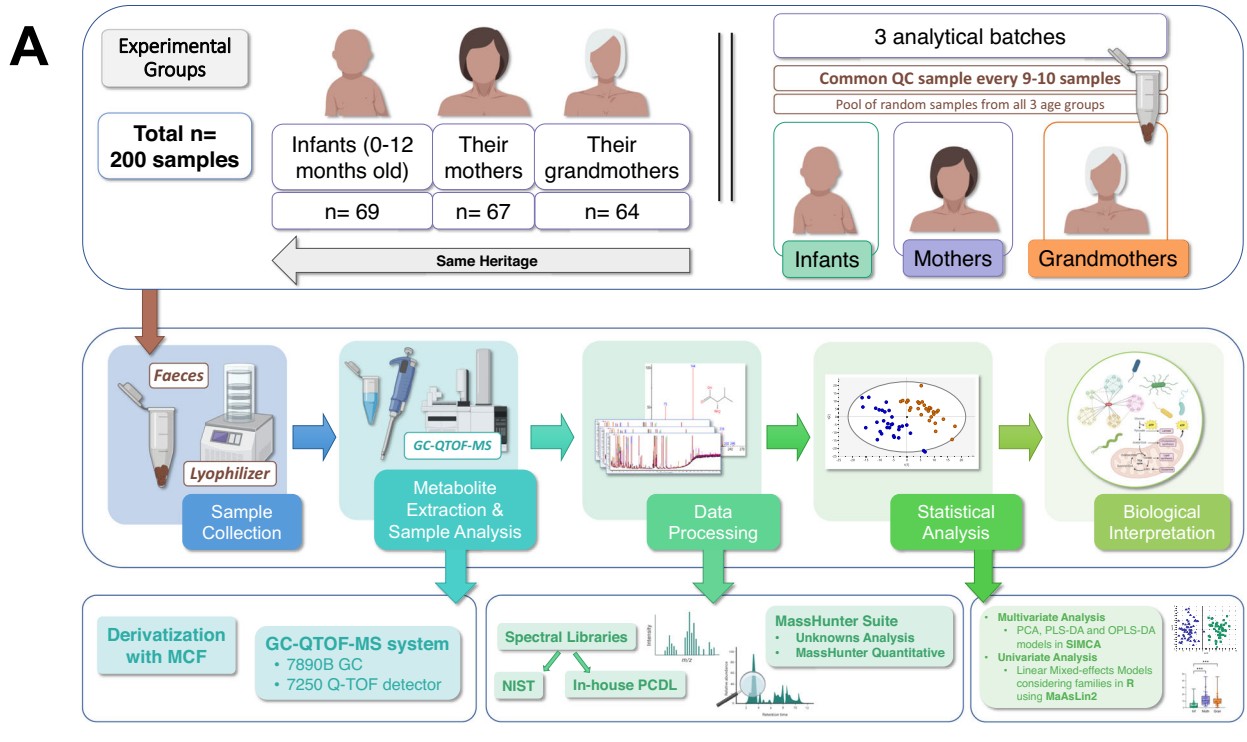

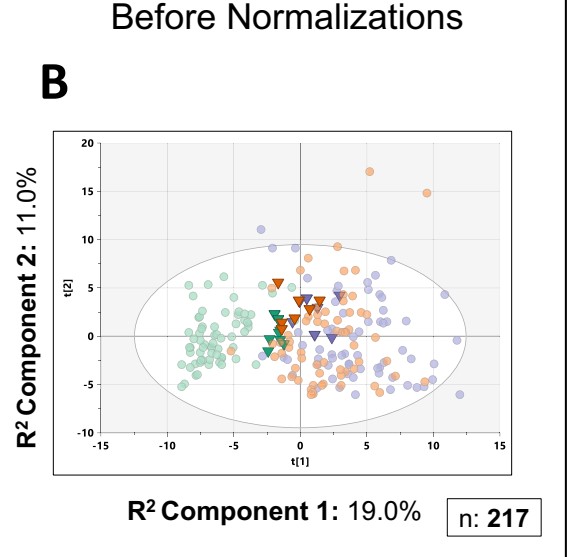

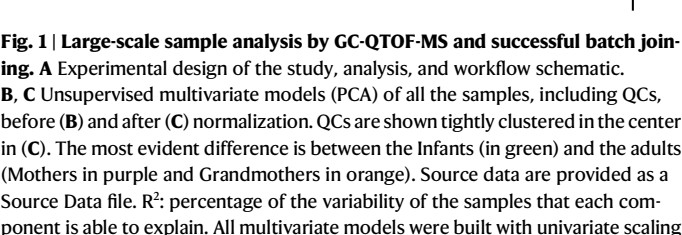

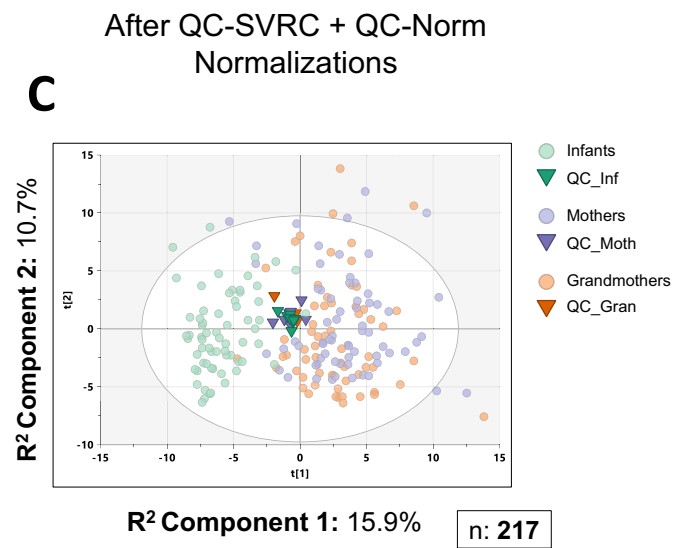

**Fig. 1 | Large-scale sample analysis by GC-QTOF-MS and successful batch joining. A** Experimental design of the study, analysis, and workflow schematic.
**B**, **C** Unsupervised multivariate models (PCA) of all the samples, including QCs, before (**B**) and after (**C**) normalization. QCs are shown tightly clustered in the center in (**C**). The most evident difference is between the Infants (in green) and the adults (Mothers in purple and Grandmothers in orange). Source data are provided as a Source Data file. R²: percentage of the variability of the samples that each component is able to explain. All multivariate models were built with univariate scaling (UV) and without any transformation of the data. Numbers of samples used to build the model (*n*) are shown in each (**B**, **C**). The full explanation for sample numbers is presented in Fig. S1. NIST: National Institute of Standards and Technology Mass Spectral Library, PCDL: Personal Compound Database and Library, QC: quality control sample, QC-Norm: inter-batch normalization by QCs, QC-SVRC: QC Sample – Support Vector Regression Correction (intra-batch normalization), PCA: Principal Component Analysis, PLSDA: Partial Least Square Discriminant Analysis, OPLSDA Orthogonal PLSDA. Icons were created using biorender.com.

Krebs cycle intermediates, indoles, and mainly fatty acids (including short- and branched-chain fatty acids, SCFA and BCFA) can be found. On the other hand, from the MSI-CE-MS data, 12 out of the 14 metabolites were significantly different between the age groups (Fig. S6B), including key metabolites that increased with age –such as butyric/isobutyric acid– and others that decreased with age –γ-aminobutyric acid (GABA), cadaverine, and choline (Supplementary Data 2).

To evaluate this data globally, all the features from both datasets –146 features from GC-QTOF-MS and 14 from MSI-CE-TOF-MS– were combined in a unified metabolomics dataset using the samples measured by both techniques (*n* = 169, see Fig. S1). This combined matrix was used to build multivariate models: firstly, a PCA model with all the samples (Fig. 2A), which showed a clear separation by the first component of the Infants, to the left, and the adults, to the right.

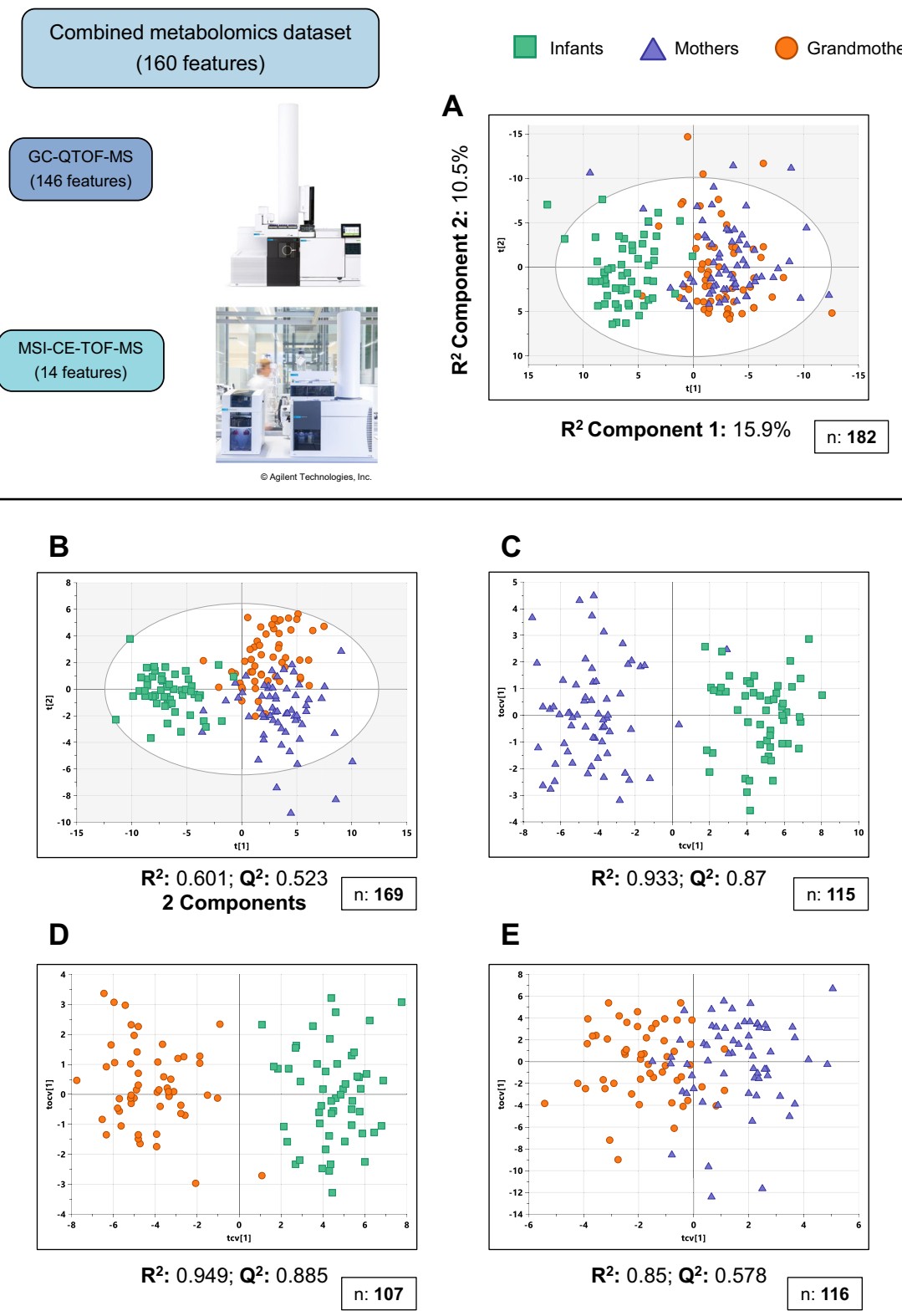

**Fig. 2 | Comparisons between age groups using the combined metabolomics dataset. A** PCA model of the combined metabolomics dataset for the samples that were measured in both techniques. **B–E** Supervised multivariate models: **B** PLS-DA model of all samples. **C** Cross-validated OPLS-DA model of Mothers *vs* Infants. **D** Cross-validated OPLS-DA model of Grandmothers *vs* Infants. **E** Cross-validated OPLS-DA model of Grandmothers *vs* Mothers. R²: percentage of the variability of the samples that the model is able to explain. Q²: capacity of the model to predict the classification of new samples. Values of R² and Q² closer to 1 indicate higher quality of the model. All multivariate models were built with univariate scaling (UV) and without any transformation of the data. Numbers of samples used to build the model (*n*) are shown in each (**A–E**). The full explanation for the differences in sample numbers and the removal of outliers is given in Materials and Methods and presented in Figs. S1 and S2. Source data are provided as a Source Data file.

This clustering of the Infants was also shown to be independent from the most important covariates: sex, type of birth, milk allergy status (allergic *vs* non-allergic), and type of diet (breastfeeding or formula) (Fig. S6C–F).

Then, we built supervised models using partial least square discriminant analysis (PLS-DA) among the three groups (Fig. 2B) and orthogonal-PLS-DA (OPLS-DA) models for pairs of groups (Fig. 2C–E). All in all, these models showed that the greatest differences were between Infants and the adult groups (Mothers and Grandmothers), where the prediction scores ($Q^2$) were close to 90% (Fig. 2C, D).

## Pathway analysis of metabolic differences between the age groups

Statistically significant and identified metabolites (FDR < 0.05, Supplementary Data 1 and 2) were used to perform a metabolic pathway enrichment analysis (Fig. 3A, Table S2). Seven pathways were significant (FDR < 0.05), including *"Biosynthesis of unsaturated fatty acids"*, *"Butanoate metabolism"*, *"Alanine, aspartate and glutamate metabolism"*, and *"Propanoate metabolism"*. Additionally, although not significant (FDR > 0.05), *"Citrate cycle"* and *"Tryptophan metabolism"* were also selected for their importance in the context of the host-microbiome co-metabolism.

Regarding the *"Biosynthesis of unsaturated fatty acids"* pathway, while the levels of saturated fatty acids (i.e., palmitic and stearic acids), monounsaturated fatty acids (MUFA; i.e., oleic acid) and linoleic acid were higher in adults, the opposite was observed for polyunsaturated fatty acids (PUFA; e.g., arachidonic and docosahexaenoic acids) (Fig. 3B). Thus, the MUFA/PUFA ratio was calculated and showed to be increased with age (Fig. 3C). On the other hand, several pathways related to energetic and biosynthetic pathways – including *"Alanine, aspartate and glutamate metabolism"* and *"Citrate cycle"* – were represented in Fig. 4A, including metabolites that are significantly higher in Infants, such as choline, glucose (and other hexoses), GABA, succinate, and citrate. Finally, in the case of *"Tryptophan metabolism"*, the levels of tryptophan presented a trend to decrease with age, along with a significant increase in its catabolic metabolites, both from host (i.e., anthranilic acid) and microbial pathways (i.e., indole and 3-methylindole/skatole) (Fig. 4B).

Boxplots for other important metabolite classes –SCFAs, BCFAs, and polyamines– were depicted in Fig. 5. Regarding SCFAs (Fig. 5A), in addition to their related pathways (*"Butanoate metabolism"* and *"Propanoate metabolism"*), Infants displayed higher levels of acetic acid and lower amounts of the other SCFAs, namely propionic, (iso)butyric, caproic, and (iso)valeric acids. Similarly, all BCFAs were significantly increased with age (Fig. 5B) and, as a result, the SCFA/BCFA ratio was higher in the Infant group (Fig. 5C). Lastly, polyamines (i.e., putrescine and cadaverine) were also shown to decrease with age (Fig. 5D). Our data demonstrated that Infants exhibit a distinctive metabolic profile that significantly differs from that of their Mothers and Grandmothers in several microbiota-derived metabolites.

## Infants show a different – and less diverse – microbial signature than their Mothers and Grandmothers

To analyze the gut microbiota, we applied metagenomic techniques (16S rRNA gene and shotgun sequencing, Fig. 6A) to a subset of the fecal samples (*N* = 128, 64.0% of the original cohort, Fig. S1A). A total of 19,523,010 sequences were generated from the 16S rRNA gene sequencing analysis. Of these, 12,955,391 remained after filtering for quality and length and removal of chimeras, resulting in 9641 amplicon sequence variants (ASVs) for the gut microbiome. Likewise, for shotgun sequencing, a total of 3,569,152,984 paired-end reads with an average of 3,379,879 sequences per sample were obtained.

Compared with Infants, adults showed a significant increase of the α-diversity (Shannon diversity index) and richness using 16S rRNA gene sequencing (*p* < 0.001) (Fig. 6B, C). This finding was further evidenced when presenting the major taxa obtained in this analysis separated by each age group, as shown in Fig. 7A, B for microbial phyla, and Fig. 7C, D for microbial genera by both 16S rRNA gene and shotgun sequencing. Extensive differences can be appreciated in most taxa regardless of the technique. Phylum relative abundances demonstrate that Infants have a much higher proportion of Actinobacteria (synonym Actinomycetota) and Proteobacteria (synonym Pseudomonadota), while Firmicutes (synonym Bacillota) are the main phylum in the adult microbiota (Fig. 7A, B). In addition, more than 50% of the Infant gut microbiota is composed of three genera – *Bifidobacterium*, *Escherichia/Shigella*, and *Veillonella* – and some that contribute highly to the Infant microbiota are almost absent in adults – e.g., *Klebsiella* and *Lactobacillus*. In contrast, other genera that are abundant in the adult microbiota are practically absent in Infants – e.g., *Faecalibacterium*, *Blautia* and *Roseburia* (Fig. 7C, D). The relative abundance of Bacteroidetes phylum (synonym Bacteroidota) and *Bacteroides* genus remains similar in both groups of age in both techniques (Fig. 7A–D). Overall, the taxonomic distribution in each age group between 16S rRNA gene and shotgun sequencing data was highly similar for microbial phyla and genera (Supplementary Data 3).

To test the significance of these differences between the groups, we applied GLM statistical analysis. Significant ASVs for phylum (Fig. S7A, B, Supplementary Data 4) and for genus (Fig. S7C, D, Supplementary Data 5) were obtained. When comparing phyla, 9 and 12 differences were found when comparing Infants individually to both, Mothers and Grandmothers for 16S rRNA gene and shotgun sequencing, respectively (Supplementary Data 4). Additionally, 75 and 85 genera were significantly different between Infants and their mothers or grandmothers according to 16S rRNA and shotgun sequencing, respectively (Supplementary Data 5). No significant differences were found between Mothers and Grandmothers in phyla nor in genera.

We further analyzed the ASVs data using the linear discriminant analysis (LDA) effect size (LEfSe) tool in the Galaxy / Hutlab webpage, which resulted in the LDA scores shown in Fig. 7E and the cladogram shown in Fig. 7F. These results confirm that the main genera enriched in the Infant microbiota – compared to the other two groups – are *Bifidobacterium* and *Lactobacillus*, while Mothers are enriched in the Clostridia class (*Clostridium III* group and *Blautia* genus), and Grandmothers have higher levels of the Deltaproteobacteria class and Verrumicrobia phylum (synonym Verrucomicrobiota).

Furthermore, for shotgun sequencing, the analysis of species was carried out (Fig. S7E). The results confirmed that the microbiota of elders was more diverse than that of the Infants, where 7 species including *Bifidobacterium bifidum*, *B. breve*, *B. longum*, *Escherichia coli*, and *Faecalibacterium prausnitzii* accounted for the 40% of the microbiota in Infants. These were confirmed to be significant after GLM analysis (Fig. S7F, Supplementary Data 6).

Additionally, the abundances of several bacterial phyla, genera, and species known to be producers of relevant metabolites were correlated with the abundances of the corresponding metabolites: GABA (Table S3) and SCFAs (Table S4). The significant results are presented in Fig. S8. Of note, the levels of GABA correlate with the abundances of GABA-producing phyla – i.e., Actinobacteria (Spearman rho, ρ = 0.240, *p* < 0.05) –, genera: *Veillonella* (ρ = 0.551, *p* < 0.001), *Bifidobacterium* (ρ = 0.362, *p* < 0.001) and *Enterococcus* (ρ = 0.453, *p* < 0.001) –, and species – i.e., Bifidobacterium *breve* (ρ = 0.382, *p* < 0.001) (Table S3 and Fig. S8A). As for SCFAs, significant correlations appeared for the genera *Bifidobacterium* (acetic acid; ρ = 0.243, *p* < 0.01), *Roseburia* (propionic acid: ρ = 0.329, *p* < 0.001), *Blautia* (propionic acid; ρ = 0.289, *p* < 0.01), *Faecalibacterium* ((iso)butyric acid; ρ = 0.526, *p* < 0.001), *Clostridium* clusters III, IV, and XIVb (valeric acid; ρ = 0.194/0.425/0.488, *p* < 0.05/*p* < 0.001/*p* < 0.001), and the species *Faecalibacterium prausnitzii* ((iso)butyric acid: ρ = 0.56, *p* < 0.001) (Table S4 and Fig. S8B).

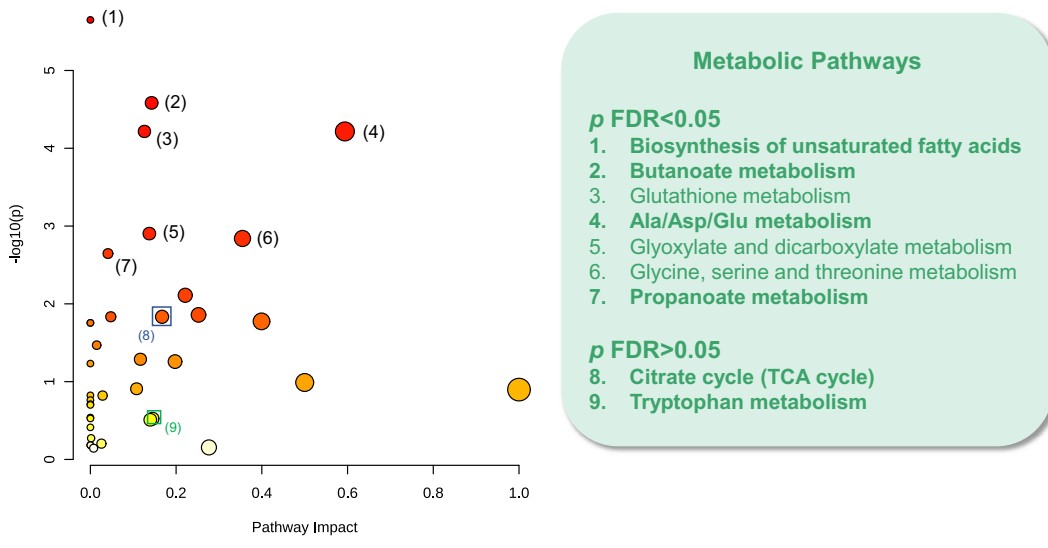

## A: Metabolic Pathway Analysis

**Metabolic Pathways**

*p* FDR<0.05
1. **Biosynthesis of unsaturated fatty acids**
2. **Butanoate metabolism**
3. Glutathione metabolism
4. **Ala/Asp/Glu metabolism**
5. Glyoxylate and dicarboxylate metabolism
6. Glycine, serine and threonine metabolism
7. **Propanoate metabolism**

*p* FDR>0.05
8. **Citrate cycle (TCA cycle)**
9. **Tryptophan metabolism**

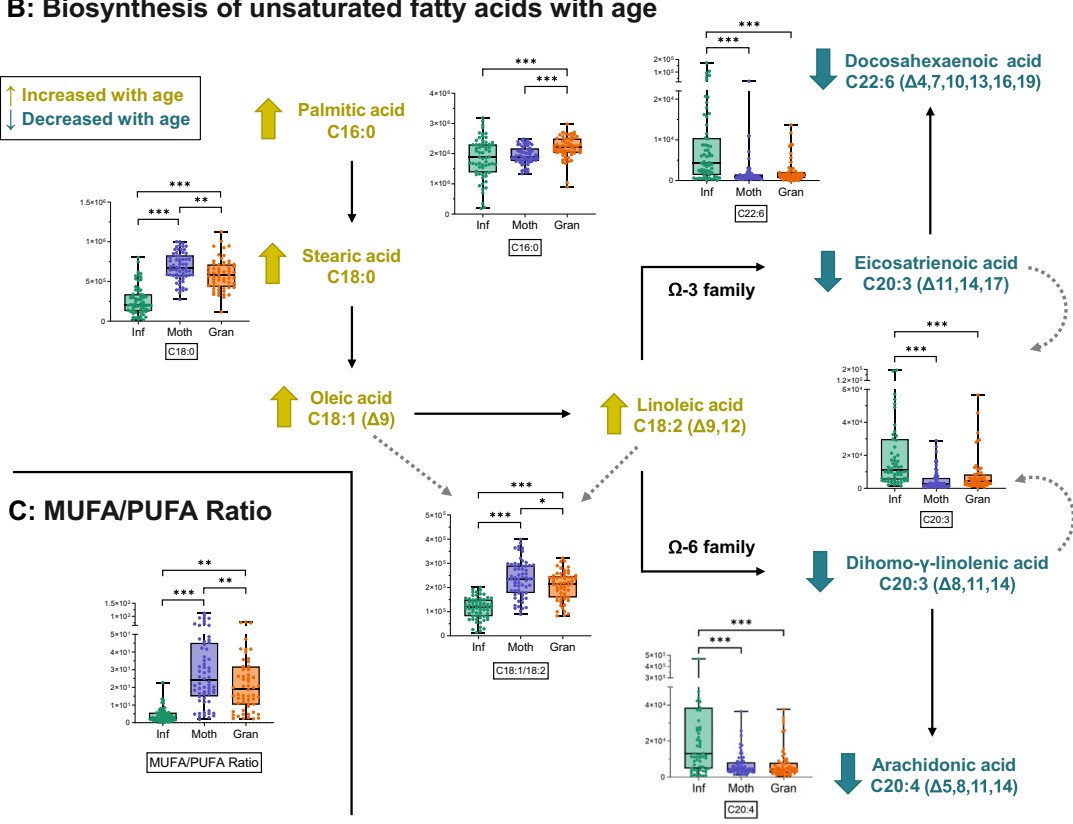

## B: Biosynthesis of unsaturated fatty acids with age

## C: MUFA/PUFA Ratio

**Fig. 3 | Metabolic pathway analysis. A** Scatter plot of the pathway analysis obtained using the MetaboAnalyst online tool v.6 (www.metaboanalyst.ca/). Parameters used: Enrichment method: Hypergeometric Test; Topology analysis: Relative-betweenness Centrality; Reference metabolome: Homo Sapiens KEGG metabolome. **B** Metabolic pathway of fatty acid metabolites related to aging. Names highlighted in gold indicate that the metabolite is increased with age, while names in blue indicate that the metabolite decreases with age. Significantly altered metabolites are presented as box and whiskers plots (statistical test was linear mixed-effects model with the correction for multiple test comparisons, FDR *p*-value < 0.05). The box and whiskers plots show the box ranging from the first to the third quartile, and the center the median value, while the whiskers extend from each quartile to the minimum or maximum values. The graph shows normalized abundance of the metabolite in the Y-axis and age group in the X-axis (*n* = 63,

*n* = 64, *n* = 58 of biologically independent samples for Infants, Mothers, and Grandmothers groups, respectively). **C** The MUFA/PUFA ratio according to age was presented as box and whiskers plots, with the box ranging from the first to the third quartile, and the center the median value, while the whiskers extend from each quartile to the minimum or maximum values. The graph shows normalized abundance of the metabolite in the Y-axis and age group in the X-axis (*n* = 63, *n* = 64, *n* = 58 of biologically independent samples for Infants, Mothers and Grandmothers groups, respectively). Statistical test was linear mixed-effects model with the correction for multiple test comparisons, FDR *p*-value < 0.05. **p* < 0.05; ***p* < 0.01; ****p* < 0.001, exact *p*-values and FDR *p*-values are *p*rovided in Supplementary Data 1. Ala: alanine, Asp: aspartic acid, Glu: glutamic acid, MUFA: mono-unsaturated fatty acids, PUFA: polyunsaturated fatty acids, TCA: tricarboxylic acids. Source data are provided as a Source Data file.

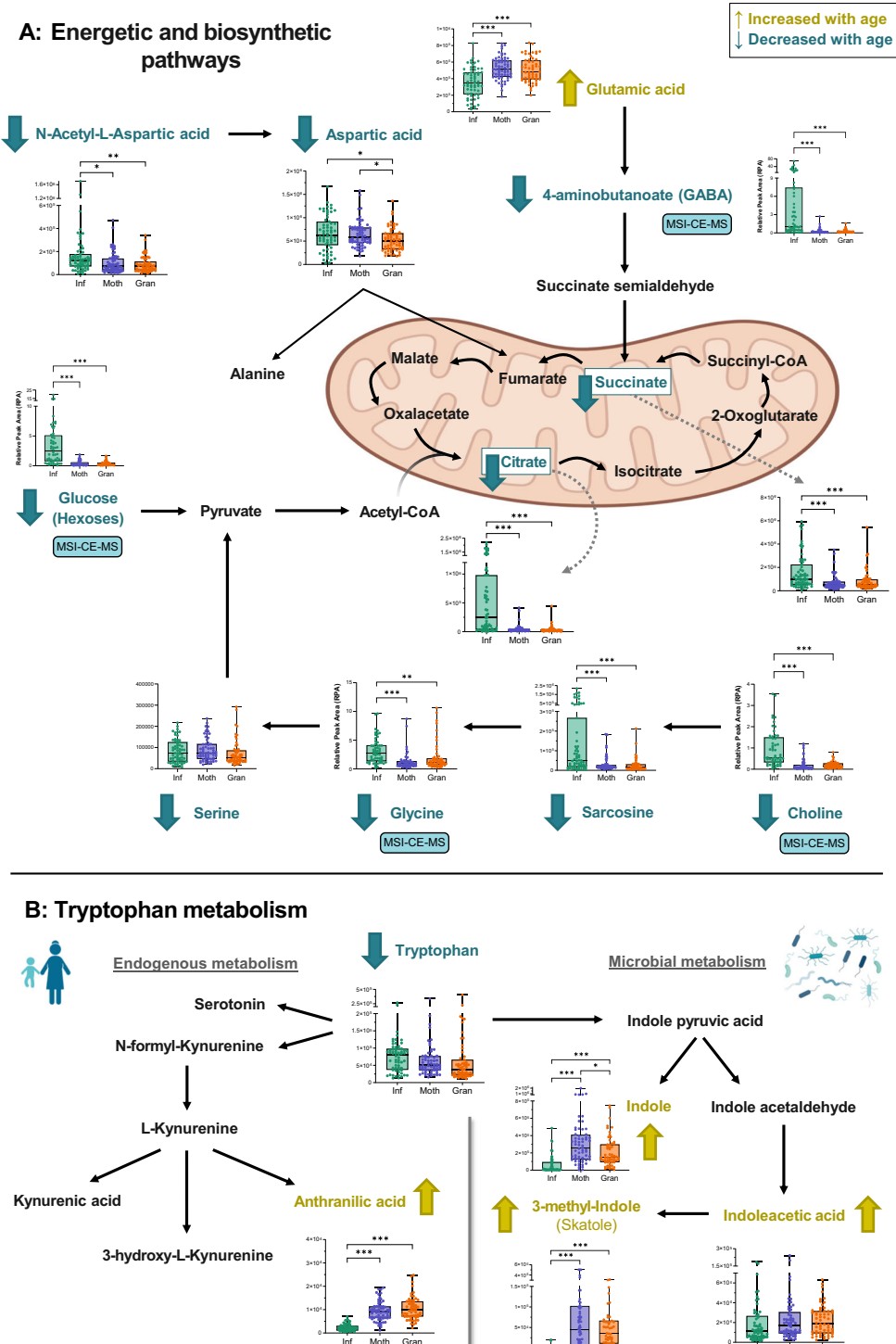

**Fig. 4 | Other metabolic pathways of interest altered by age. A** Pathway related to energetic metabolism showing significantly altered metabolites using box and whiskers plots (statistical test was linear mixed-effects model with the correction for multiple test comparisons, FDR *p*-value < 0.05). In this, the box ranges from the first to the third quartile, and the center the median value, while the whiskers extend from each quartile to the minimum or maximum values. The graph shows normalized abundance of the metabolite in the Y-axis and age group in the X-axis. All metabolites were from GC-QTOF-MS (*n* = 63, *n* = 64, *n* = 58 of biologically independent samples for Infants, Mothers, and Grandmothers groups, respectively), except when explicitly noted as from MSI-CE-MS (*n* = 54, *n* = 62, *n* = 55 of biologically independent samples for Infants, Mothers and Grandmothers groups, respectively).
**B** Pathways related to endogenous and microbial metabolism of tryptophan showing significantly altered metabolites using box and whiskers plots (statistical test was

linear mixed-effects model with the correction for multiple test comparisons, FDR *p*-value < 0.05). In this, the box ranges from the first to the third quartile, and the center the median value, while the whiskers extend from each quartile to the minimum or maximum values. The graph shows normalized abundance of the metabolite in the Y-axis and age group in the X-axis. All metabolites were from GC-QTOF-MS (*n* = 63, *n* = 64, *n* = 58 of biologically independent samples for Infants, Mothers and Grandmothers groups, respectively), except when explicitly noted as from MSI-CE-MS (*n* = 54, *n* = 62, *n* = 55 of biologically independent samples for Infants, Mothers and Grandmothers groups, respectively). For A and B pathways, names highlighted in gold indicate that the metabolite is increased with age, while names in blue indicate that the metabolite decreases with age. *p* < 0.05; **p* < 0.01; ***p* < 0.001, exact *p*-values and FDR *p*-values are provided in Supplementary Data 1 and 2. Source data are provided as a Source Data file. Icons were created using biorender.com.

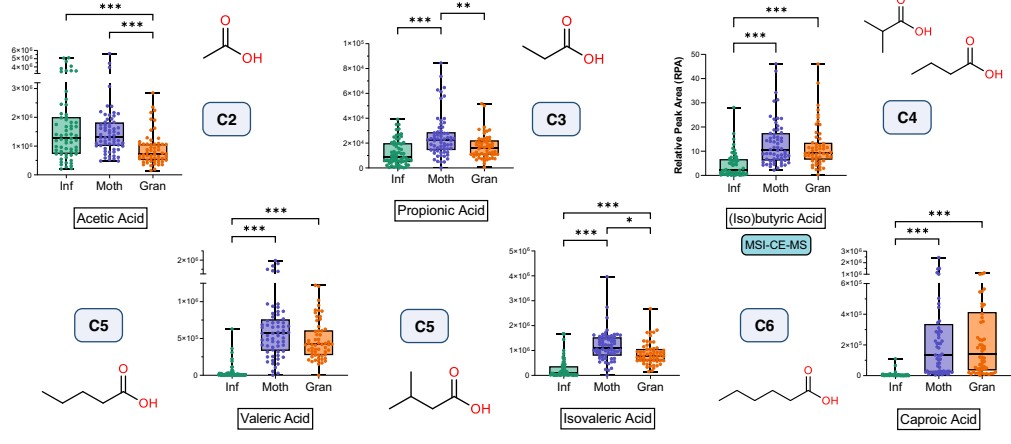

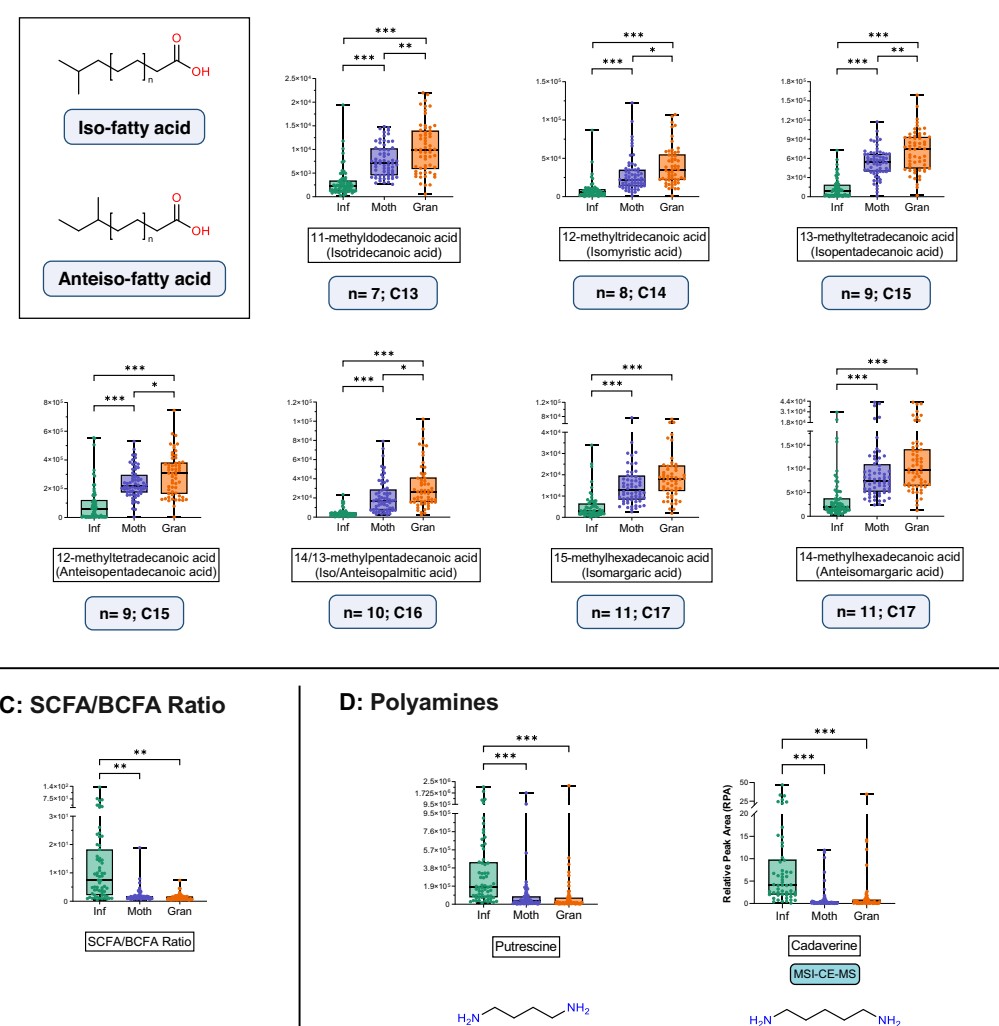

**Fig. 5 | Trajectories of other metabolite classes of interest altered by age.**
**A** Short-chain fatty acids (SCFAs). **B** Branched-Chain Fatty Acids (BCFAs). **C** The SCFA/BCFA ratio. **D** Polyamines. The metabolites in (**A**–**D**) are presented as box and whiskers plots, with the box ranging from the first to the third quartile, and the center the median value, while the whiskers extend from each quartile to the minimum or maximum values. The graph shows normalized abundance of the metabolite in the Y-axis and age group in the X-axis. All metabolites were from GC-QTOF-MS ($n = 63$, $n = 64$, $n = 58$ of biologically independent samples for Infants,

Mothers, and Grandmothers groups, respectively), except when explicitly noted as from MSI-CE-MS ($n = 54$, $n = 62$, $n = 55$ of biologically independent samples for Infants, Mothers, and Grandmothers groups, respectively). Statistical test was linear mixed-effects model with the correction for multiple test comparisons, FDR $p$-value < 0.05. *$p < 0.05$; **$p < 0.01$; ***$p < 0.001$, exact $p$-values and FDR $p$-values are provided in Supplementary Data 1 and 2. Source data are provided as a Source Data file.

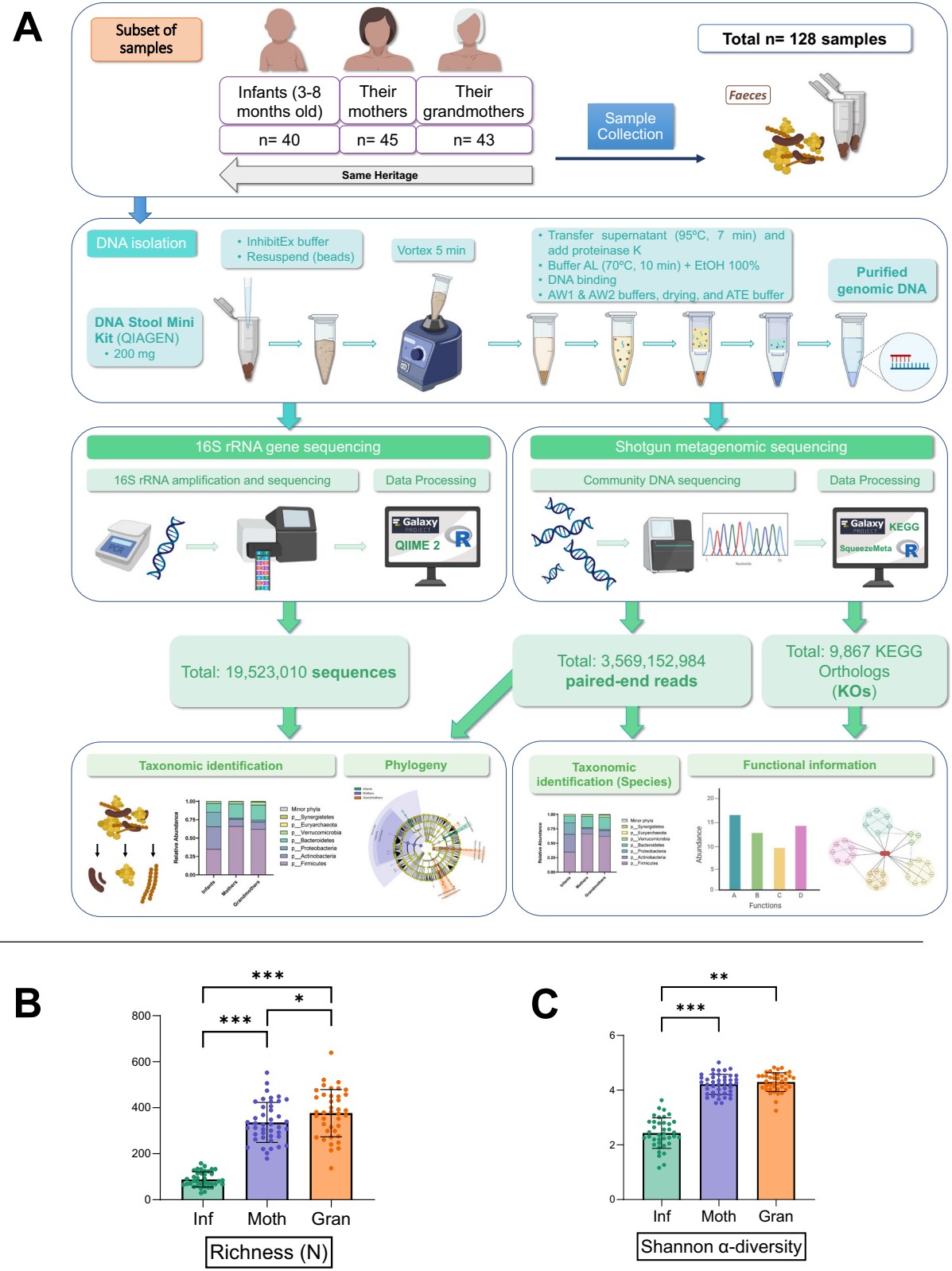

**Fig. 6 | Genomics analysis (16S rRNA gene sequencing and shotgun metagenomics) of a subset of samples. A** Experimental design of the study, analysis, and workflow schematic. **B** Richness parameter of the three age groups ($n = 38$, $n = 43$, $n = 40$ of biologically independent samples for Infants, Mothers, and Grandmothers groups, respectively). Data are presented as mean values ± SD; statistical test was linear mixed-effects model, $p$-value < 0.05. **C** Shannon α-diversity of the three age groups ($n = 38$, $n = 43$, $n = 40$ of biologically independent samples for Infants, Mothers, and Grandmothers groups, respectively). Data are presented as mean values ± SD; Statistical test was linear mixed-effects model $p$-value < 0.05. *$p$ < 0.05; **$p$ < 0.01; ***$p$ < 0.001. Source data are provided as a Source Data file. Icons were created using biorender.com.

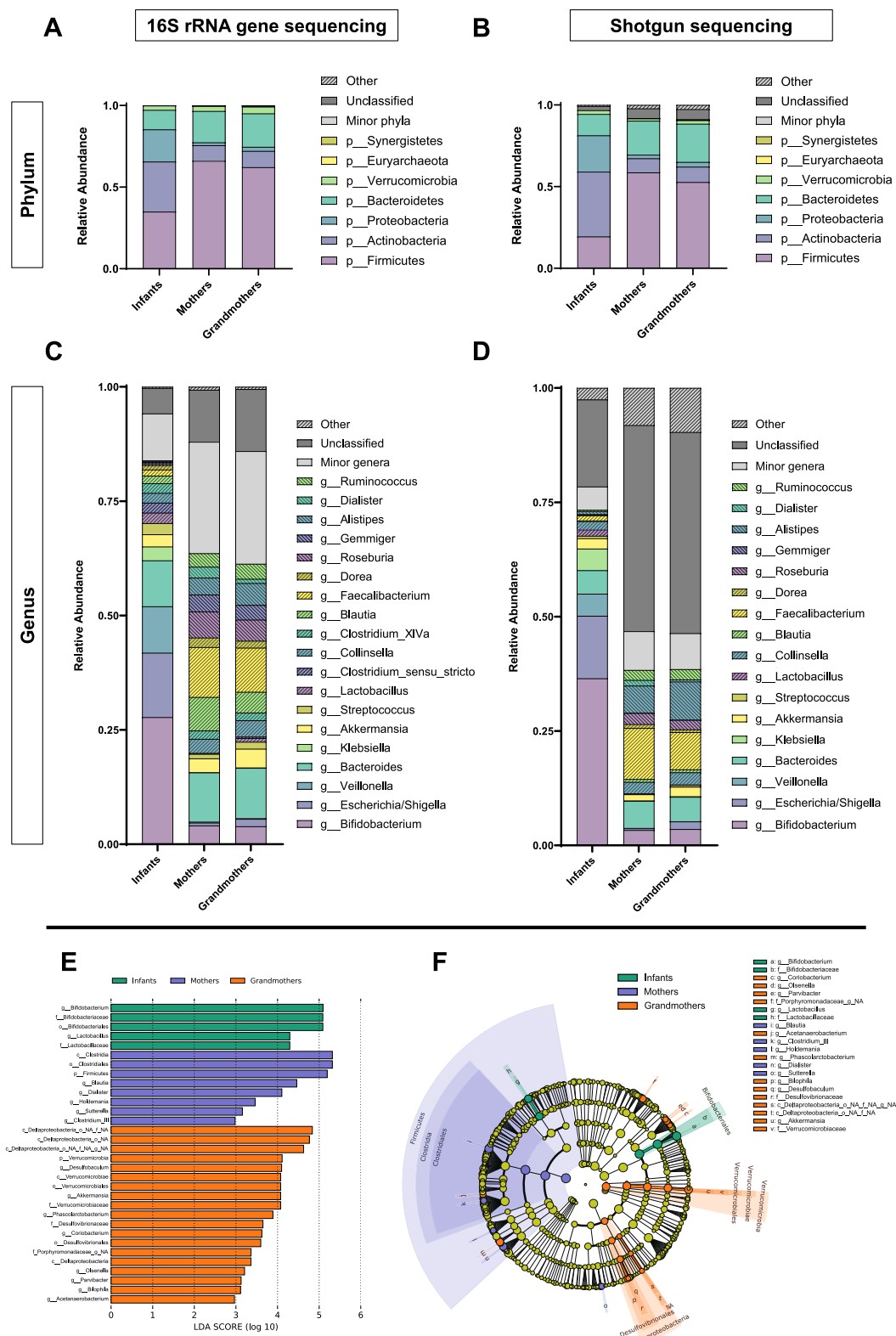

**Fig. 7 | Taxonomic differences between generations according to 16S rRNA gene and shotgun sequencing data. A**, **B** Main phyla for each age group using 16S rRNA gene and shotgun sequencing, respectively. **C**, **D** Main genera for each age group using 16S rRNA gene and shotgun sequencing, respectively. Data are represented using stacked bars. Minor phyla: <0.1% relative abundance in all groups. Minor genera: <2% relative abundance in all groups. Other: Unmatched phyla/genera with minor relative abundance. **E** Linear discriminant analysis (LDA) scores resulting from the LDA-effect size (LEfSe) tool in the Galaxy / Hutlab web-page. **F** Cladogram of the results of the previous LEfSe analysis, showing the taxa that are enriched in each age group, categorized by their taxonomic relationships. Source data are provided as a Source Data file.

These results show that the Infants harbor a much less diverse microbiota, mainly composed of *Bifidobacterium*, *Escherichia/Shigella*, and *Veillonella*, and they present extensive differences with both Mothers and Grandmothers in many taxa. On the other hand, the two groups of adult women do not significantly differ, suggesting that the changes in the microbiota of adults are more subtle.

## The functionality of the Infant microbiota differs in many pathways from that of adults

A total of 9867 KEGG Orthologs (KOs) were obtained from our metagenomic data after applying SqueezeMeta, of which 4708 became significant between Infants and adults (Mothers or Grandmothers) upon statistical analysis (Supplementary Data 7). These significant KOs were grouped according to their respective KEGG Pathways and tested between the age groups at two different levels: BRITE level B (corresponding to pathways such as *"Energy metabolism"*; Supplementary Data 8) and level C (corresponding to more specific pathways, such as *"Glycolysis/Gluconeogenesis"*; Supplementary Data 9). Results from level B showed that many general pathways were significantly different between Infants and adults, including *"Carbohydrate metabolism"* and *"Energy metabolism"* (Fig. 8A). Further analysis into level C pathways confirmed this, with pathways such as *"Glycolysis / Gluconeogenesis"*, *"Pentose phosphate pathway"* and *"Starch and sucrose metabolism"* being more abundant in Infants whereas *"Oxidative phosphorylation"* and *"Carbon fixation pathways in prokaryotes"* were enriched in the adults (Supplementary Data 9). These pathways are reflective of the type of diet and metabolic capacity, which is highly different between Infants and adults.

Among the significant annotated KOs, functions related to the metabolism of fatty acids (i.e., K11533), SCFAs (i.e., K00135, K04072, K01902, K01903) and citrate cycle (i.e., tricarboxylic acids, TCA cycle: K00239, K00240, K01902, K01903) were more abundant in Infants. On the other hand, polyamines signaling (i.e., K01596, K00850) and BCFAs metabolism (i.e., K00248, K00249) associated entries were increased with age. Lastly, for tryptophan metabolism, opposite trajectories were observed for enzymatic activities participating in its biosynthesis (i.e., K01696, TrpB, reduced with age) and catabolism (i.e., K01667, TnaA, increased with age) (Fig. 8B, Supplementary Data 7).

## Integration of different omics data – metabolites, ASVs, and KOs – using mixOmics demonstrates a high correlation among them and reveals the unique signature of the Infants

With the aim of integrating the three omics datasets – (i), metabolites from both GC-QTOF-MS and MSI-CE-MS, (ii) ASVs from 16S rRNA gene sequencing, and (iii) KOs from shotgun metagenomics –, we applied the mixOmics package using the *Data Integration Analysis for Biomarker discovery using Latent Components* (DIABLO) framework[46,47]. Selection of ASVs taxonomic data from 16S rRNA gene over shotgun sequencing was due (i) to high correlation for the 85 most abundant matched genera (with a minimum abundance of at least 1000 counts among samples) showed by 90% of them being significant and all positively correlated ($\rho > 0.70$ in 70% of the genera, Table S5); and (ii) to include information from three independent datasets.

Thus, we used the data from the samples that were measured successfully in all the analyses ($N = 113$, see Fig. S1B and Supplementary Data 10 for explanations). The results of the script are displayed in Fig. 9. Sample plots (PLS-DA models) of the DIABLO model in Fig. 9A displayed that the three omics datasets were able to discriminate between Infants and adults, in agreement with what was described in the previous sections separately. Furthermore, receiver operating characteristic curves (ROC) showed that the comparison "Infants *vs* Others" was the one with the highest Area Under the Curve (AUC) in all three datasets (>0.9), being the KOs dataset the most discriminating with an AUC value of 0.9929 (Fig. 9B). Figure 9C showed that the latent

components of each omic dataset were highly correlated between each other, highlighting that DIABLO could model a good agreement between the datasets. The best correlation was between KOs and ASVs with an index of 0.94. Once again, the scatterplots show the Infants clearly separated from the adults in their own cluster. The strongest correlations between the most important features of each dataset are presented in a circosPlot (Fig. 9D) which displays the different types of significantly correlated features on a circle and shows their link between omics or within an omic, indicating strong positive or strong negative correlations (in red and blue, respectively). Almost all strong correlations are positive, except for glucose and *Bifidobacterium* which are negatively correlated with most KOs from this subset. Lastly, the unique multi-omic signature of the Infants was depicted using a clustered image map (CIM) (Fig. 9E). Hierarchical clustering of the samples using the 75 best features from the model (25 from each dataset) classified the Infants almost perfectly into their own cluster, with only one Grandmother wrongly classified as an Infant. Mothers and Grandmothers, as before, did not separate well into distinct clusters. The signature (the rows) is clearly visible, with only the 9 variables in the top section being higher in the Infants, while the other 66 are lower. From these 9 variables higher in the Infants, we can highlight previously mentioned genera (*Bifidobacterium*, *Veillonella*, and *Escherichia/Shigella*) and metabolites (glucose, putrescine, and choline). On the other hand, among others, *Faecalibacterium*, linoleic/oleic acid, and several BCFAs (isotridecanoic acid, isopentadecanoic acid, and iso/anteisopalmitic acid), appear as the most important variables that are lower in the Infant population. Of note, all the KOs cluster together and show a specific signature, distinct from the other two omic datasets. Finally, this information was represented and integrated in a pathway analysis using Cytoscape (Fig. S9).

## Discussion

The crosstalk mechanisms between the microbiota and host metabolism are complex, and how they change with age is still not fully understood. To our knowledge, this is a pioneering study characterizing the fecal microbiome and metabolome of Infants, their Mothers, and Grandmothers, leaving out heterogeneity due to the genetic line. The samples were analyzed using two omics methodologies: metabolomics (using a novel GC-QTOF-MS protocol in an untargeted approach and MSI-CE-MS in a targeted approach) and two genomic approaches (16S RNA gene sequencing and shotgun metagenomics).

GC-MS is a crucial technique that has been hampered in its use for large-scale analyses. Combining a cutting-edge derivatization technique[31] with a batch strategy and the use of a common QC sample, we have demonstrated that this methodology can be applied successfully to the analysis of large-scale studies. The GC-QTOF-MS analysis was complemented by the MSI-CE-MS protocol which allowed us to detect key metabolites of the host-gut microbiota crosstalk and round up our biological interpretation.

Among the results from metabolomics, the changes in MUFA and PUFA have been studied in relation to aging through their role in oxidative stress[48]. This is because, while MUFAs are practically resistant to this phenomenon, PUFAs are more susceptible to lipid peroxidation via reactive oxygen species (ROS) due to the presence of multiple double bonds[49]. Thus, the decrease of PUFA levels (C20:3, C22:6, and C20:4) in adults compared with the Infants could be due to higher levels of lipid peroxidation, wherein PUFAs are metabolized into oxidized PUFAs that act as free radicals and contribute to cellular aging. In contrast, MUFA (e.g., C18:1) have been seen to prolong lifespan[50], and therefore could be considered as potent regulators of longevity. In line with this, a recent study showed that increased levels of MUFA and decreased levels of PUFA in a young mouse model inoculated with aged microbiota correlated with alterations in the expression of different genes related to lipid biosynthesis[51]. Of note, here we observed alterations in different bacterial functional activities regarding fatty

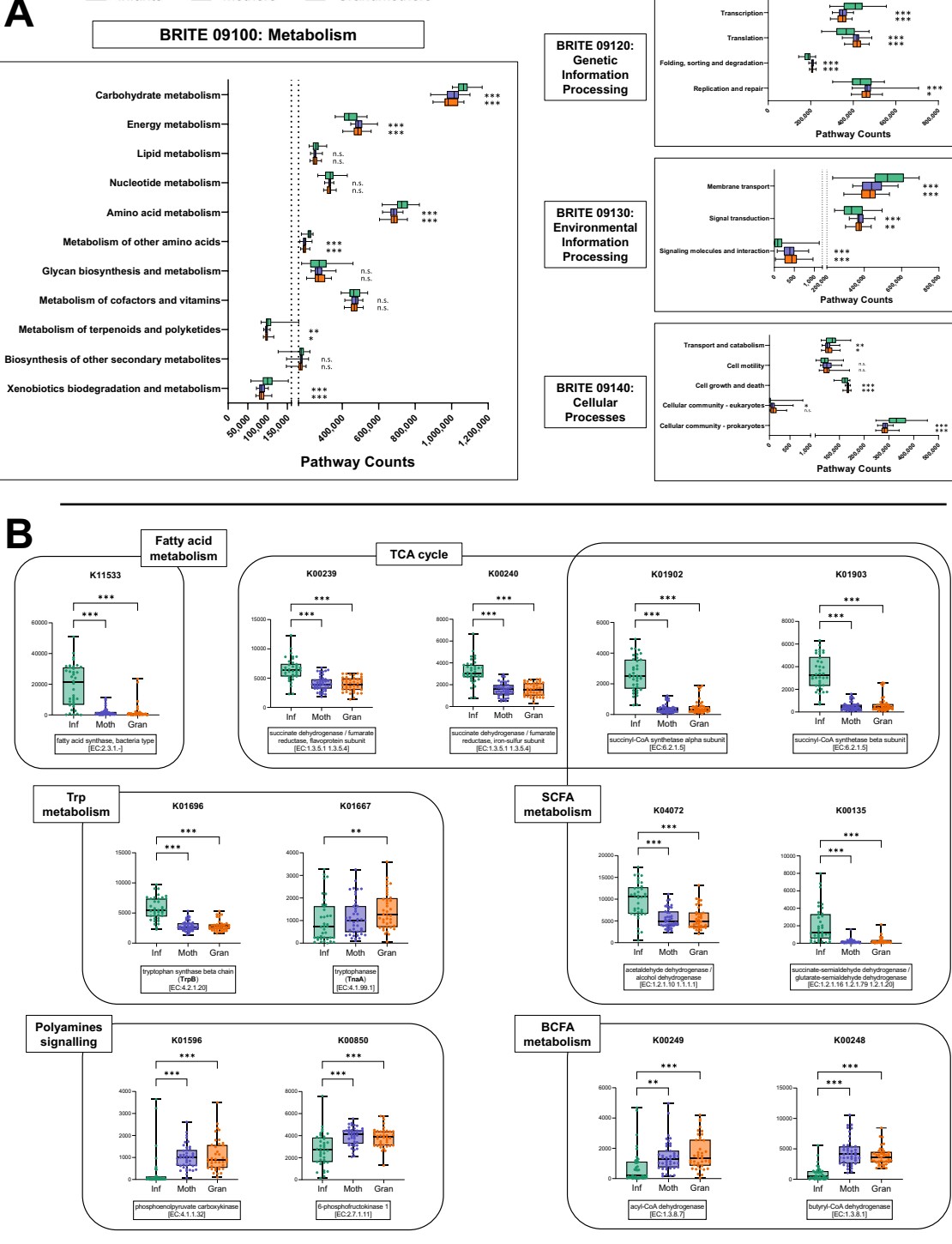

**Fig. 8 | Functional pathways in the three generations according to shotgun metagenomics. A** Functional pathways from level B of the BRITE Hierarchies, grouped by their level A ("Metabolism", "Genetic Information Processing", "Environmental Information Processing" and "Cellular Processes"). Significant pathways are presented as box and whiskers plots (statistical test was linear mixed-effects model with the correction for multiple test comparisons, FDR *p*-value < 0.05). For the box and whiskers plots, the box ranged from the first to the third quartile, and the center the median value, while the whiskers extend from each quartile to the minimum or maximum values (*n* = 38, *n* = 43, *n* = 40 of biologically independent samples for Infants, Mothers, and Grandmothers groups, respectively). The graphs show pathway counts in the X axis. Complementary information is presented in Supplementary Data 8 for level B and Supplementary Data 9 for level C pathways.

**B** Selected KEGG Orthologues (KOs) grouped by pathways of interest are presented as box and whiskers plots (statistical test was linear mixed-effects model with the correction for multiple test comparisons, FDR *p*-value < 0.05). For the box and whiskers plots, the box ranged from the first to the third quartile, and the center the median value, while the whiskers extend from each quartile to the minimum or maximum values (*n* = 38, *n* = 43, *n* = 40 of biologically independent samples for Infants, Mothers and Grandmothers groups, respectively). Complementary information is presented in Supplementary Data 7. EC: enzyme commission number. Source data are provided as a Source Data file. Significance: *\*p* < 0.05; \*\**p* < 0.01; \*\*\**p* < 0.001; n.s.: not significant; *\*p* < 0.05; \*\**p* < 0.01; \*\*\**p* < 0.001, exact *p*-values and FDR *p*-values are provided in Supplementary Data 7, 8, and 9.

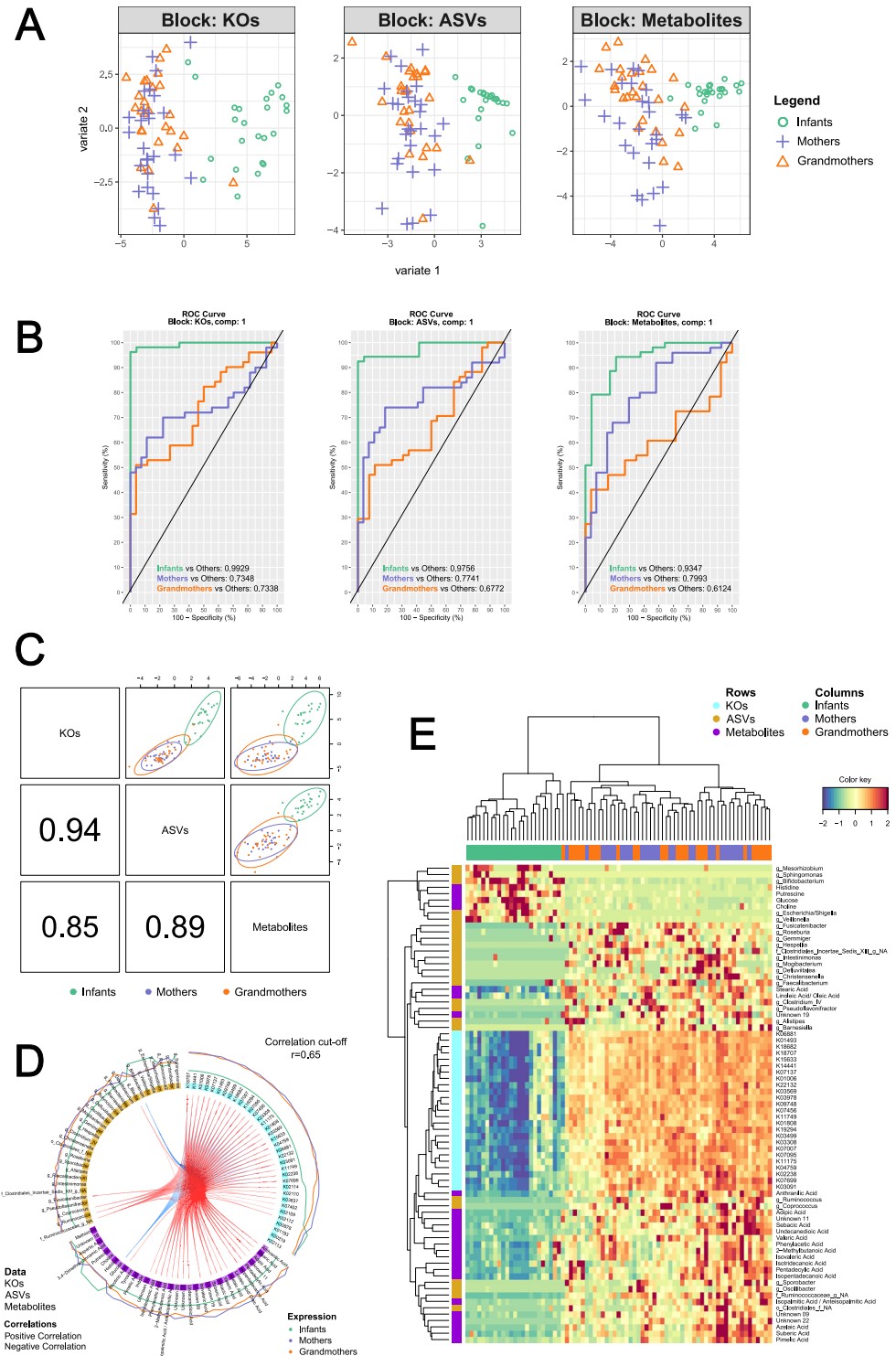

**Fig. 9 | Integration of the three omics using mixOmics and DIABLO. A** Sample plots (PLS-DA models) of the final DIABLO model for each omic dataset. **B** ROC Curves for each dataset. Values of the Area Under the Curve (AUC) of each "one-*vs*-all" comparison are superimposed on top of the corresponding graph. **C** Sample scatterplot from plotDiablo (the graphical output for the DIABLO framework) displaying the first component in each data set (upper diagonal plot) and Pearson correlation between each component (lower diagonal plot). Samples are represented with a 95% confidence ellipse. The bottom left numbers indicate the correlation coefficients between the first components from each data set. **D** circosPlot of the most significant variables from each dataset and their pairwise correlations. Variables are represented on the side of the circosPlot, where colors indicate the type of data (KOs, ASVs or Metabolites), and external lines display the measured levels with respect to each outcome category (Infants, Mothers or Grandmothers). In the center, red lines signify a positive correlation between the linked variables, and blue lines a negative one. **E** Clustered Image Map (CIM) from the cimDiablo function showing the multi-omic molecular signature expression of the samples (in columns) for the 75 most important variables from the model (top 25 from each dataset, in rows). A hierarchical clustering – built with Euclidean distance and complete linkage method – was applied to both rows and columns. This CIM shows the clear differential multi-omic signature of the Infants, who are perfectly clustered. The legend of KOs from Heatmap and circosplot is shown in Table S6. Source data are provided as a Source Data file.

acid synthesis pathways, such as fatty acid synthase type II (FAS-II, K11533), greatly decreased in adults.

In addition, our metabolomics and metagenomics data suggest that there is an inverse and possibly gut microbiome-mediated relationship between age and several energetic and biosynthetic pathways, including the TCA cycle. The age-related decline in mitochondrial function, where the TCA cycle takes place, has already been described and has been attributed to several causes[52]. However, very little is known about the influence of the microbiota on this process, but we believe our results could shed some light in this matter. Mitochondrial dysfunction is a hallmark of human aging, showing loss of electron chain oxygen consumption and ATP production[53]. Furthermore, several studies have linked alterations in the microbiota to various pathologies involving mitochondrial dysfunction[54–57] and TCA alterations[51,58], suggesting a crosstalk between the microbiota and mitochondria. In this respect, metabolites such as GABA may have a direct impact on energetics by controlling the availability of substrates such as glutamate[41,59]. Indeed, the higher levels of GABA in Infants when compared with both adult groups correlate with high abundances of GABA-producing phyla (Actinobacteria), genera (*Veillonella, Bifidobacterium* and *Enterococcus*), and species (*Bifidobacterium bifidum, B. breve, B. longum*) – the latter one according to the Virtual Metabolic Human database[60] – as shown in Table S3 and Fig. S7A.

Regarding tryptophan, this essential amino acid can be metabolized via endogenous or bacterial pathways[61]. Because it depends on a microbiota that changes over the years, tryptophan metabolism is also affected by age[62]. Interestingly, our metagenomic data showed opposite trajectories for enzymatic activities of its biosynthesis and catabolism: TrpB (which catalyzes the last two steps in the biosynthesis of Trp) decreased with age, while TnaA (tryptophanase, a bacterial enzyme involved in its degradation to indole) increased with age. These results, together with lower levels of tryptophan, suggest there is an upregulation of tryptophan catabolism with increasing age[2]. At the biological level, both tryptophan and its metabolites act as aryl hydrocarbon receptor (AhR) ligands and regulate the expression of genes involved in the modulation of inflammation and intestinal homeostasis[61]. Consequently, tryptophan has been considered a potent regulator of lifespan and age-related diseases like inflammaging[63].

In addition, SCFAs have key roles as energy sources[48] and as signaling molecules that contribute to intestinal homeostasis[64–67]. They are the main metabolites produced by the human microbiota through the anaerobic fermentation of undigested proteins and carbohydrates in the colon[68,69]. In the case of Infants, their diet is based on milk (whether breastmilk, formula or a combination of both), which is mainly composed of oligosaccharides that are metabolized by different strains of *Bifidobacterium*[70]. The higher levels of *Bifidobacterium* (genera and species) in Infants correlated positively with acetic acid, the shortest SCFA ($\rho = 0.21$ to $0.25$, $p < 0.05$). On the contrary, the composition of the adult microbiota was characterized by a higher bacterial richness according to Shannon α-diversity. Adult microbiota was more abundant in Bacteroidetes and Firmicutes phyla, SCFA-producing genera, such as *Faecalibacterium, Roseburia, Blautia*, and *Clostridium* clusters, and species such as *Faecalibacterium prausnitzii*. Consequently, adults showed increased levels of longer-chain SCFAs (specifically propionic, butyric/isobutyric acid, valeric, isovaleric, and caproic acids)[71,72]. As shown in the boxplots, the differences are higher the longer the carbon chain of the SCFA. These correlations were proven statistically, as shown before, for *Roseburia, Blautia, Clostridium* clusters III, IV, and XIVb, *Faecalibacterium* genus and *Faecalibacterium prausnitzii* species (Table S4 and Fig. S8B)[73].

Regarding BCFAs, our data showed that their levels increased with age which, in addition to the longer-chain SCFA data, proves the negative correlation between proteolytic and saccharolytic capacity with age[74]. BCFAs are methylated saturated fatty acids derived from several sources which are considered to promote health effects[75,76]. In humans, they are believed to derive mainly from the diet (especially from ruminant-derived products, such as milk, cheese or beef) and they are present to a lesser extent in several tissues including skin, adipose tissue, and serum[76,77]. Thus, their higher levels in adults may be a direct consequence of a diet with higher quantities of ruminant-derived products. In addition, BCFAs are major components of bacterial membranes across many genera and species[78], so they arise as well from bacterial activity. Finally, they are products of the endogenous catabolism of branched-chain amino acids (BCAAs), which are three essential amino acids – valine, leucine, and isoleucine – that are believed to be involved, among others, in the regulation of different biological processes including aging[79–81]. Our study suggests that BCAA conversion to BCFA is altered with age, as enzymatic functions from this catabolic pathway (e.g., K00248, K00249) were more abundant in adults than Infants.

Lastly, polyamines such as putrescine or cadaverine are essential molecules for cell proliferation that come from the diet and endogenous or microbial decarboxylation of amino acids[82]. It has been shown that levels of polyamines decrease with age and that their administration increases lifespan in animal models[83]. Surprisingly, the same effect was achieved only by administrating probiotic *Bifidobacterium* strains, giving rise to the importance of an Actinobacteria-enriched microbiota, such as that of Infants, in the production of polyamines[84]. These polycationic metabolites may exert a longevity effect through different mechanisms, including AMP-activated protein kinase (AMPK) activation[48]. In support of this, our study found that microbial enzymatic activities related to the AMPK pathway (i.e., phosphoenolpyruvate carboxykinase and 6-phosphofructokinase-1) were increased in adults.

The three omics techniques individually raised complementary information about the same metabolic pathways, allowing us an integrative view of how the interaction between the host and the microbiota changes over the lifespan. Although each omic technique showed good discrimination parameters, it was only the integration of the three different datasets using mixOmics that allowed to draw an almost perfect fingerprint for the microbiota of Infants. This was possible as high correlation in the most abundant taxa between 16S rRNA gene and shotgun sequencing was demonstrated, as stated by other authors[85,86]. Thus, in our mixOmics results, the functions (represented by the KOs) seem to be the most discriminant characteristic among them. This, coupled to the fact that the highest correlation parameters were found between KOs and ASVs (that come from different analyses), showed that not only is the microbiota of Infants different, but it also behaves differently and is accountable for the changes that we see between age groups.

The most significant features presented in the CIM and the circosPlot figures belonged, among others, to the metabolic pathways that we had previously identified in our metabolomics, 16S rRNA gene sequencing, and shotgun metagenomics data. Interestingly, the top 35 KOs were mainly increased in adults and belonged, among others, to pathways related to the carbohydrate and energy metabolism categories from KEGG Orthology BRITE Database (e.g. subunits β (K02109), γ (K02110), and ε (K02114) from bacterial F-type ATPase). Additionally, most of these KOs negatively correlated with the most characteristic features of the gut microbiota of Infants (*Bifidobacterium* and glucose), proving that the most discriminant difference between Infants and adults is the energetic functionality of their microbiota, which is directly related to their type of diet. In the case of Infants, their diet based on milk gears their microbiota towards a saccharolytic metabolism (as evidenced by metabolites such as glucose, and their overexpressed metagenomic pathways), carried out mainly by the genera– *Bifidobacterium, Escherichia/Shigella*, and *Veillonella*, and the species – *B. bifidum, B. breve, B. longum*, and *E.coli* – that conform more than 50%

and 40%, respectively, of their microbiota. In contrast, these genera account for less than 7% of the adult microbiota, which is much more diverse, and this is reflected in their metabolic profile with the higher abundances of SCFAs and BCFAs and lower levels of sugars and TCA metabolites.

The strengths of this one-of-a-kind study include the sample collection, including three different generations of participants from two hospitals and five health centers in Madrid; the broad epidemiologic variables obtained, and the rigorous and state-of-the-art analysis in two omics methodologies and 4 high throughput techniques. However, this study had certain limitations which should be considered in future studies. The stool samples were collected at a single time point in the study; therefore, it was not possible to assess changes in the composition of the microbiota over time, provide information on evolution, or monitor the threshold at which the microbiome becomes more stable and similar to that of adults. In addition, the dietary variable was highly confounded with the age groups, as it is closely linked to the circumstances of each age period, making it difficult to dissociate the effects of diet and aging.

All in all, the integration of our data showed that the microbiota of infants is metabolically, compositionally, and functionally distinct from that of adults. Our study stands out as the pioneer to integrate different omics approaches in an intergenerational study while controlling genetic variability and provides valuable data that could be used as a starting point for new hypotheses. It is also worth noticing that human samples were used instead of animal models, and the pathways were reported in detail –including the bacterial enzymes involved as well as the actual metabolites– providing compelling evidence of the interrelated processes and significantly strengthening the biological interpretation. In addition, the results have been presented in a clear and visually appealing manner, and we believe they represent a significant contribution to the field of microbiota and aging.

In any case, further studies are needed to elucidate the relationship between these processes and how they can be improved for human health, especially by monitoring the same participants over time and providing prospective follow-up data on birth outcomes later in life. In addition, other sample matrices –such as the serum metabolome–, and other metabolomics platforms –such as lipidomics– may provide valuable information and complement the results of this work in the future.

## Methods

### Study design

We performed an intergenerational and observational study approved by the Regional Ethics Committee for Clinical Research of *Hospital Universitario Infantil Niño Jesús* in Madrid according to the ethical guidelines outlined in the Declaration of Helsinki and its amendments[87]. The cohort of participants was initially recruited for a study regarding cow's milk allergy, which showed scarce differences between allergic and non-allergic infants[88]. Participants were recruited at *Hospital Universitario Infantil Niño Jesús*, *Hospital General Universitario Gregorio Marañón* and five Health Centers in Madrid (Spain). All participants provided informed consent.

### Study population

The final study population consisted of 200 participants: 69 Infants up to twelve months of age, their 67 Mothers, and their 64 maternal Grandmothers. Infants were sex balanced, with a slightly higher proportion of females (55%) (Table S1). None of the infants had transitioned into a solid diet, and all were fed with either breastmilk alone, formula milk, or a combination of both. Mothers and grandmothers reported no dietary restrictions or specific diets. Exclusion criteria for all age groups were antibiotics intake in the 3 months prior to study recruitment and any other concomitant severe disease. Due to the

difficulties in locating families with all three generations willing to participate, recruitment of the research population took two years.

All participating centers applied the same recruitment protocols and questionnaires. Information on age, sex, mode of birth, antibiotics use at birth, cow's milk allergy and feeding regime for Infants, and age and smoking habits of Mothers and Grandmothers were collected. This data is presented in Supplementary Data 10, along with the family number and the techniques and datasets in which each sample was measured and/or included.

### Sample collection and processing

Each participant received a sample collection kit developed by our group for easier collection at home. We published this procedure elsewhere[88,89]. Subjects were asked to record the sample collection date, to not contaminate the sample with urine or toilet paper and to store the samples in the home freezer at −20 °C before taking them to the laboratory. A total of 200 fecal samples were collected at the laboratory and stored at −80 °C before being processed for metabolomics analysis and DNA extraction.

### Gas chromatography coupled to QTOF mass spectrometry analysis (GC-QTOF-MS)

Prior to this study, we adapted a derivatization method that allows for shorter derivatization times <15 min[31] compared to the 16 h of traditional approaches[32,35]. It also enhances the coverage of key metabolites of the host-gut microbiota co-metabolism, such as SCFA, which have poor recoveries in the traditional derivatization reactions since they require anhydrous media and samples must be evaporated up to total dryness. We applied this methodology to a QTOF mass analyzer, a high-resolution equipment well-known for its powerful capabilities for unveiling unknown compounds. It allows for the determination of accurate mass in all detected mass-to-charge (*m/z*) fragments, which enormously enhances the selectivity and the sensitivity of the analysis and leads to a much more reliable identification[32,90]. The samples analyzed were $n = 67$, $n = 67$, $n = 63$ of biologically independent samples for Infants, Mothers, and Grandmothers groups, respectively. The number of QCs were $n = 24$, and the biological replicates were $n = 1$.

**Sample preparation.** The protocol for sample preparation was adapted from Zhao et al.[31], with some modifications[42]. Briefly, feces samples were lyophilized overnight at −110 °C in a LyoQest −85 lyophilizer (Telstar) until completely dried. Then, the lyophilizate from each individual sample was homogenized mechanically. Samples were taken as quickly as possible from the −80 °C freezer to avoid any thawing that would interfere with the lyophilization.

**Metabolite extraction.** 10 mg of lyophilized feces were mixed with 300 μL of NaOH (1 M) solution in Milli-Q water and 10 μL of 1000 ppm 4-methyl valeric acid (4MV) solution, IS1, in a 0.5 mL Eppendorf tube and vortexed vigorously for 2 min. Samples were then centrifuged at $16,000 \times g$ at 4 °C for 20 min. Each 200 μL of supernatant was transferred into an autosampler vial, and the residue was further extracted with 200 μL of cold methanol. After repeating the homogenization and centrifugation as above, 167 μL of supernatant was combined with the first supernatant in the sample vial. Blank samples were prepared in the same way, omitting the 10 mg of lyophilizate.

**Derivatization with methylchloroformate (MCF).** To derivatize the extracted metabolites, 20 μL of MCF were added to each vial, as well as 34 μL of pyridine. The samples were then vortexed vigorously for exactly 30 s. Another 20 μL of MCF were added again, and samples were vortexed for another 30 s. Subsequently, 400 μL of 20 ppm tricosane (TRI) solution, IS2, in chloroform were added, and samples were vortexed for 30 s, followed by an addition of 400 μL of sodium bicarbonate solution (NaHCO$_3$, 50 mM in Milli-Q water) and additional

vortexing for 30 s. Samples were then centrifuged at $2000 \times g$ for 20 min at 4 °C in order to clearly visualize the double meniscus. Finally, 350 μL of the bottom chloroformic phase were transferred to transparent GC vials containing ~100 mg of anhydrous sodium sulfate ($Na_2SO_4$).

**GC-quadrupole-time of flight (QTOF)-MS equipment and analysis.** The GC system (Agilent Technologies 7890B) consisted of an autosampler (Agilent Technologies 7693 A) and an Accurate-Mass Q-TOF (Agilent Technologies 7250). Data collection was performed with MassHunter Workstation GC/MS Data Acquisition Version 10.0. (Agilent).

GC-QTOF parameters were optimized from Zhao et al.[31] to increase the coverage of relevant SCFAs by enabling the analysis of the most volatile SCFAs. One μL of each derivatized sample was injected using a splitless injection technique into a DB-5 MS capillary column (30 m × 0.25 mm i.d., 0.25 μm film thickness; (95% dimethyl/5% diphenyl) polysiloxane bonded and cross-linked with a precolumn (10 m) integrated; Agilent J&W Scientific, Folsom, CA), with helium as the carrier gas at a constant flow rate of approximately 1.0 mL·min⁻¹. The solvent delay time was set to 2.9 min. The optimized temperature gradient was the following: 30 °C held for 1 min, then increased at a rate of 10 °C·min⁻¹ up to 120 °C for 9 min, 20 °C·min⁻¹ to 320 °C for 10 min, and then held there for 2 min. The total time of analysis was 22 min. The temperatures of the injector, transfer interface, and ion source were set to 270, 270, and 220 °C, respectively. Electron impact ionization (70 eV) at the examined $m/z$ range of 38 − 650 was used. The acquisition rate was 20 spectra·s⁻¹. To avoid signals from the solvents that could saturate the MS, the detector was turned off in the following retention time (RT) segments: 3.52–3.90 and 4.84–5.30.

**Experimental design for large-scale sample analysis in GC-QTOF-MS: batches and QC strategy.** To be able to measure reliably all the samples from our cohort we divided them into three batches, according to the age group – batch 1: 67 Mothers, batch 2: 63 Grandmothers, and batch 3: 67 Infants. The samples were randomly measured inside their batch.

To later be able to join these batches and compare the data between them, we devised a quality control (QC) strategy. To prepare the QC samples, 17 mg of 30 lyophilized samples were taken (10 random samples from each of the 3 age groups) and a pool of 510 mg was obtained. This pool was split into three parts, one for each batch (170 mg per batch). Inside each batch, 11 QC samples were prepared, by taking 10 mg of the lyophilized feces and following the same protocol as the samples. From these 11 QCs, 3 were used as equilibrium QCs at the start of the analysis, and the other 8 were placed in the worklist every 9–10 samples – whose order was randomized inside each batch. This assured that the batches could be joined later by using a normalization strategy. Schematics detailing the preparation of QC samples and the worklists measured in the equipment are presented in Fig. S10.

Additionally, for GC-QTOF-MS a mix of n-alkanes was injected at the beginning of the worklist to later build calibration files that would allow converting RT into retention index (RI). This parameter is later used for the enhancement of metabolite annotation, as explained below.

**Data treatment and compound identification in GC-QTOF-MS.** The data treatment procedure was adapted from recently published protocols that have been rigorously tested and fine-tuned at CEMBIO[32]. GC-QTOF-MS data, peak detection, and spectra processing algorithms were applied using MassHunter Software (Agilent) using an untargeted workflow.

The overall analytical performance was carefully examined by inspection of total ion chromatograms (TIC) of experimental samples,

QC samples, blanks, and both internal standards (IS): 4MV and TRI, following the guidelines of previous work on QA[91]. 4MV is the IS1 that is added before the derivatization and thus controls the reproducibility of the whole sample preparation process, while TRI is the IS2, added after the derivatization, and is used to control mostly the analytical performance of the GC-QTOF-MS equipment (drifts in response and performance of the injection process).

Then, the SureMass mass spectral deconvolution algorithm was applied in MassHunter Unknowns Analysis (B.10.0, Agilent) to identify co-eluted compounds by matching deconvoluted spectra and/or RTs to two spectral libraries: (i) a "Personal Compound Database and Library" (PCDL) built by analyzing chemical standards with the same method and (ii) a further search into the NIST20 library (National Institute of Standards and Technology, U.S. Department of Commerce). The details and parameters of these analyses are described in detail elsewhere[32].

For obtaining the abundances of the metabolites, MassHunter Quantitative (B.10.0, Agilent) was used. First, the target ions were carefully checked with the experimental data, avoiding saturated ions and discarding compounds derived from the solvents, derivatizing agents or column bleeding. This process allowed the better revision of compound annotation using the NIST20 library with the tool MS Interpreter in NIST MS Search v.2.4 in MassHunter Qualitative (B.08.00, Agilent).

The result of the peak integration was an Excel file (Microsoft® Excel® for Microsoft 365 MSO v. 2401) for each batch (Infants, Mothers, and Grandmothers) ($N = 161$ features). The following step was a blank subtraction to ensure that signals from the reagents, solvents, contaminants, and derivatizing agents were removed ($N = 152$). Then, a quality assurance procedure (QA) was performed[92,93]. First, we applied a filter by presence, where features that appeared in <50% of QCs and <70% of samples were removed ($N = 152$). Then, we applied a filter by Relative Standard Deviation (RSD), where metabolites were kept if they had, in at least one batch, an RSD < 30% in the QC samples ($N = 134$). This was performed independently for each batch.

Normalization procedures were also carried out to ensure the comparison between the three batches. First, an intra-batch correction using "QC Sample – Support Vector Regression Correction" (QC-SVRC)[43], and after the fusion of the three batches in a unique Excel, an inter-batch correction with QC-Norm[44,45] were applied. After these normalizations, the QA filters resulted in 152 features after the presence check and 146 final features after the RSD filter.

**SCFA/BCFA and MUFA/PUFA ratios.** The SCFA/BCFA ratio was calculated as the sum of the abundances of (Acetic + Propionic + Valeric + Caproic acid) divided by the sum of (Isotridecanoic acid + Isomyristic acid + Isopentadecanoic acid + Anteisopentadecanoic acid + [Iso/Anteiso]palmitic acid + Isomargaric Acid + Anteisomargaric acid).

The MUFA/PUFA ratio was calculated as the abundance of "Linoleic Acid/Oleic Acid" divided by the sum of (Arachidonic Acid + Eicosatrienoic Acid + Docosahexaenoic Acid). Both ratios were calculated from the data of GC-QTOF-MS for these metabolites.

**Multivariate statistical analyses.** After all the procedures described above, the data were imported into SIMCA (v.16.0.1, Sartorius Stedim Data Analytics AB) to build multivariate statistics models. Non-supervised models (PCA) and supervised models (PLS-DA and OPLS-DA) were obtained. The cross-validated scores from the OPLS-DA models and the "Variable Importance for the Projection" (VIP) values were also obtained from the significant models.

**Multisegment injection (MSI) capillary electrophoresis coupled to TOF mass spectrometry analysis (MSI-CE-TOF-MS)**
The samples were also measured in capillary electrophoresis coupled to MS (CE-MS), using a methodology published by Shanmuganathan

and co-authors[38]. This methodology, called Multisegment Injection (MSI)-CE-MS, is specially designed for large-scale studies, as in the present case, and employs serial injections of 13 samples in the CE system to allow for greater sample throughput (≈4 min/sample). In this case, we applied a targeted search for important metabolites of the gut microbiota metabolism that had not been previously detected in the GC-QTOF-MS analysis. The samples analyzed were $n = 57$, $n = 66$, $n = 60$ of biologically independent samples for Infants, Mothers, and Grandmothers groups, respectively. The number of QCs were $n = 31$, and the biological replicates were $n = 1$.

**Sample preparation.** We employed fecal extracts prepared by taking 25 mg of feces lyophilizate and extracting with a mixture of $H_2O$:ACN:IPA in a 10:6:5 (vol/vol/vol) proportion. This extract was filtered through Micro centrifuge filters (0.45 μm Nylon, Costar) at $16,100 \times g$ for 15 min at 4 °C. For this extract to be suitable for MSI-CE-MS analysis, 100 μl aliquots of each sample were dried in a $N_2$ evaporator and reconstituted with 50 μl of $H_2O$:Methanol in a 70:30 proportion, containing the corresponding IS – 40 μM chloro-tyrosine (Cl-Tyr) as the IS for positive mode and 1 mM napthalene monosulfonic acid (NMS) as the IS for negative mode.

QC samples were prepared by pooling 5 μl of each sample in the study to a "Grand Pool", from which six 50 μl aliquots were prepared by the same protocol as the rest of the samples. These QC samples were measured throughout the run to monitor analytical variability and equipment performance.

**MSI-CE-TOF-MS equipment and analysis.** Samples were measured in a G7100A CE equipment coupled to a 6230 TOF detector[38]. The sample preparation and analysis in the equipment were performed in the Britz-McKibbin lab, at McMaster University (Hamilton, ON, Canada). Data collection was performed with MassHunter Workstation LC/MS Data Acquisition for 6200 series TOF (Agilent).

**Targeted search of metabolites using in-house library.** Once the data were obtained, they were processed following the published procedures[38]. The QA procedures and statistical analyses were applied in the same way as for the GC-QTOF-MS data. Of note, GABA was not detected in all samples (total $n = 160$) due to isomeric interferences. Thus, a K-nearest neighbors (KNN) missing value imputation was carried out with a MATLAB script[94].

### 16S rRNA gene sequencing and shotgun metagenomic sequencing

For both genomic techniques – 16S rRNA gene sequencing and shotgun metagenomics – a subset of samples was selected, due to limitations in sample quantity. Therefore, 40 Infants, 45 Mothers, and 43 Grandmothers ($n = 128$) were included.

**DNA extraction.** DNA was extracted from 200 mg of feces using the QIAamp DNA Stool Mini Kit (QIAGEN, Hilden, Germany) following the manufacturer's instructions plus an additional membrane disruption step using metal beads[95]. The extracted DNA was quantified by Qubit® 2.0 Fluorometer following the dsDNA HS Assay Kit protocol. All 128 samples had a concentration higher than 0.2 μg/μL and were suitable for sequencing.

**16S rRNA gene amplification and sequencing.** The V3-V4 regions of the 16S rRNA gene were amplified and sequenced using the MiSeq platform from Illumina, as described in the manual for "16S Metagenomic Sequencing Library Preparation" of the MiSeq platform (Illumina, San Diego, California, EEUU).

**Bioinformatics and statistical analyses for 16S rRNA gene sequencing.** 16S rRNA sequences were denoised and processed with DADA2

v1.11 in order to define ASVs[96]. In addition, DADA2 and the command removeBimeraDenovo were used to remove chimeras. Taxonomy was assigned using the DADA2 implementation of the RDP classifier[97], using the formatted RDP training set 16 release 11.5 from the DADA2 website. Shannon's α-diversity index was estimated using the package vegan v2.6-4 and the R 4.2.2 software. Richness was defined as the number of ASVs identified in a given sample.

For the cladogram of differences in species abundance between groups, we applied the linear discriminant analysis (LDA) effect size (LEfSe) tool[98], via the interactive webpage (https://huttenhower.sph.harvard.edu/galaxy/). This high-dimensional data analysis tool for biological indicators compares various groups, focuses on biological correlations and statistical significance, and identifies biological indicators that differ significantly between groups.

**Shotgun metagenomics sequencing.** Metagenomic sequencing was performed using a NextSeq platform (Illumina, USA) according to the manufacturer's recommendations for obtaining paired-end sequences of 150 base pairs.

**16S rRNA gene and shotgun sequencing phylum taxonomy.** For both 16S rRNA gene sequencing and shotgun metagenomics sequencing, the phylum nomenclature was established using up to two nomenclature codes: previous informal names, and the new names published by the International Code of Nomenclature of Prokaryotes (ICNP)[99].

**Bioinformatics and statistical analyses for shotgun metagenomics sequencing.** First, the paired-end Fastq files were filtered by quality with the Fastp application (version 0.20.1)[100]: low-quality bases were trimmed from the tail of each read, low complex reads were removed, and reads shorter than 35 bases were discarded. Next, all the reads passing the previous filters were mapped onto the *Homo sapiens* genome (GCA_000001405.28 GRCh38.p13 assembly) by means of Bowtie2 (version 2.4.2)[101]. Finally, the reads which did not align concordantly against the *Homo sapiens* genome were analyzed with the SqueezeMeta pipeline with coassembly mode (version 1.3.1; database built on September 2020)[102]. Tabular outputs were generated from the SqueezeMeta results using the sqm2tables.py SqueezeMeta script, ignoring unclassified reads in the TPM calculation.

Next, for shotgun sequencing data, we analyzed both taxonomic data (phyla, genera, and species) and metabolic functions. For functional pathways, we tested these differences at several levels: KEGG Orthology (KO) terms, that identify orthologous groups of genes organized using the BRITE functional hierarchy[103], and level B and C of the hierarchy, explained below.

We joined these KO terms according to the BRITE functional hierarchies (see https://www.genome.jp/kegg/kegg3b.html) to which they belong, obtaining: 52 BRITE functional hierarchies (level B), (e.g., Amino acid metabolism, Carbohydrate metabolism, Endocrine system), and 483 BRITE functional hierarchies (level C), (e.g., Histidine metabolism, Galactose metabolism, Glucagon signaling pathway).

For KEGG and BRITE the following parameters were used: minimum abundance and prevalence applied were: 0.0. The normalization was performed with the trimmed mean of M values method (TMM) also using a linear model test (LM) with a maximum significance 0.10.

**Statistical univariate analysis.** Analysis was performed using the MaAsLin2 (Microbiome Multivariable Associations with Linear Models) R package[104] applying GLM models establishing the family as the random effect and the generation as the fixed effect (Infants, Mothers, and Grandmothers). The False Discovery Rate (FDR) was performed, and statistical significance was set at 95% level (FDR < 0.05). We applied this

test for the metabolomics data (GC-QTOF-MS and MSI-CE-MS), the taxonomic data from metagenomics (16S rRNA gene and shotgun sequencing) i.e. phyla, genera, and species, and the functional data from shotgun sequencing (KEGG and BRITE B and C functional hierarchies).

The MetaboAnalyst online tool (v. 6.0) was used for metabolomic pathway analysis. GraphPad Prism (v.9.5.0) was used to represent statistical graphs.

**Multi-omics dataset integration using mixOmics.** Integration of the different Omic datasets was performed with the mixOmics script[46] using the *Data Integration Analysis for Biomarker discovery using Latent Components* (DIABLO) framework[47] using R (v. 4.2.2). This is a multi-omics integrative method that seeks common information across different data types through the selection of a subset of molecular features, while discriminating between multiple phenotypic groups. For this integration we used the 113 samples that were common to all techniques used (Fig. S1B, Supplementary Data 10). We assessed the performance of the model using 5-fold cross-validation repeated 10 times with the function perf(). We then assessed the accuracy of the prediction on the left-out samples (KOs, ASVs, metabolites). ROC curves and AUC plots per block were plotted using the function auroc(). We set the number of variables to select to 25 on each of the 3 components (KOs, ASVs, Metabolites) of PLS-DA. The circosPlot was built based on a similarity matrix. A cutoff of 0.65 was included to visualize correlation coefficients above this threshold in the multi-omics signature.

### Reporting summary

Further information on research design is available in the Nature Portfolio Reporting Summary linked to this article.

## Data availability

The 16S rRNA and shotgun metagenomics sequencing files and metadata data generated in this study have been deposited in the Sequence Read Archive (SRA) at NIH database under accession code PRJNA991269. The metabolomics raw data generated in this study have been deposited to the EMBL-EBI MetaboLights database[105] under accession code MTBLS7670. In addition, the processed omics data (16S rRNA and shotgun metagenomics sequencing, and metabolomics from GC-QTOF-MS and MSI-CE-MS platforms) in this study are provided in the Source Data file. Source data are provided with this paper.

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

## Acknowledgements

We would like to thank all institutions and hospitals involved: Institute of Applied Molecular Medicine (IMMA, San Pablo CEU University, Madrid), Centre for Metabolomics and Bioanalysis (CEMBIO, San Pablo CEU University, Madrid), Spanish National Cancer Research Centre (CNIO), Hospital Infantil Niño Jesús, Instituto de Investigación Sanitaria-La Princesa, Hospital General Universitario Gregorio Marañón, Centro de Salud (C.S) Eloy Gonzalo (Feliciano López, Ana María López Madrazo, Sonia Luna Ramírez), C.S Baviera (Mercedes Velez García-Nieto), C.S Ibiza (Alberto Morlan Sala), C.S Justicia (Mª Rosario Antón Jiménez), C.S Valleaguado (Paloma Ortiz Ramos) and C.S. Goya (Yolanda Martín Peinador), Fundació per al Foment de la Investigació Sanitària i Biomèdica de la Comunitat Valenciana (FISABIO), CIBER en Epidemiología y Salud Pública, Joint Research Unit in Genomics and Health, Fundació per al Foment de la Investigació Sanitària i Biomèdica de la Comunitat Valenciana (FISABIO),

Institut de Biologia Integrativa de Sistemes (Universitat de València / Consejo Superior de Investigaciones Científicas) and McMaster University. The authors would like to thank Prof. Santiago Angulo for his statistics advice and MATLAB script development, and Clarissa Caroli for her help in acquiring the GC-QTOF-MS data. T.C.B-T would like to personally thank Ritchie Ly and Zachary Kroezen for their help in obtaining and processing the MSI-CE-MS data. Figures were created with biorender.com. This work was supported by ISCIII (PI17/01087), cofounded by FEDER "Investing in your future" for the thematic network and co-operative research centers RICORS "Red de Enfermedades Inflamatorias (REI)" (RD21/0002/0008). This work was also supported by Agencia Estatal de Investigación, Ministry of Science and Innovation in Spain (PCI2018-092930) co-funded by the European program ERA HDHL - Nutrition & the Epigenome, project Dietary Intervention in Food Allergy: Microbiome, Epigenetic and Metabolomic interactions DIFAMEM. This research was also funded by the Ministry of Science and Innovation of Spain (MCIN) MCIN/AEI/10.13039/501100011033, by ERDF A way of making Europe, grant number PID2021-122490NB-I00. P.B.-M. acknowledges financial support from the Natural Sciences and Engineering Research Council of Canada, Genome Canada, and the Canada Foundation for Innovation. A.V. acknowledges financial support from "Programa Ayudas Puente 2023–2024" from San Pablo CEU University Foundation and grant number PEJ-2023-AI/SAL-GL-27622 from the Community of Madrid for hiring a research assistant. T.C.B-T. and A.M-C. are supported by an FPI-CEU predoctoral fellowship and T.C.B-T. acknowledges the CEINDO – BANCO SANTANDER fellowship for his international research mobility in Canada. D.B. acknowledges financial support from Instituto de Salud Carlos III (PI19/00044). M.P-G. and E.Z-V. acknowledge financial support from Instituto de Salud Carlos III (PI20/01366).

## Author contributions

M.P-G. and M.D.I-S. were the PIs of the project, secured funding, and sample collection to writing. A.V., D.B. and C.B. also secured funding, supervised all the stages of the project and its conceptualization. E.Z-V., R.B., P.C-F., M.P-G. and M.D.I-S. recruited participants and collected and categorized the samples. T.C.B-T. and E.Z-V. obtained the GC-QTOF-MS data using a methodology developed by M.F.R-S., who also supervised the analysis, together with C.B. and A.V. T.C.B-T. performed the data treatment under the supervision of M.F.R-S., C.B. and A.V. T.C.B-T. obtained the MSI-CE-TOF-MS data and analyzed it under the supervision of M.S. and P.B-M. E.Z-V., L.M-B. and I.A.M-A. obtained the genomics data in FISABIO under the supervision of M.P-G., C.U. and M.P.F. E.Z-V. and L.M-B. performed the data treatment of the 16S rRNA sequencing data, and L.A. and I.A.M-A. performed the data treatment of the shotgun metagenomics data and the multi-omics integration. A.M-C. researched the literature and constructed the biological interpretation under the supervision of T.C.B-T., E.Z-V., M.P-G. and A.V. T.C.B-T., E.Z-V. and A.M-C. created the figures and tables of the manuscript. The first draft was written by T.C.B-T., E.Z-V., A.M-C., M.P-G. and A.V. Comments on the manuscript and approval of the final version were provided by all the authors.

## Competing interests

The authors declare no competing interests.

## Additional information

[1]Centro de Metabolómica y Bioanálisis (CEMBIO), Facultad de Farmacia, Universidad San Pablo-CEU, CEU Universities, Boadilla del Monte, Spain. [2]Departamento de Ciencias Médicas Básicas, Instituto de Medicina Molecular Aplicada (IMMA) Nemesio Díez, Facultad de Medicina, Universidad San Pablo-CEU, CEU Universities, Boadilla del Monte, Spain. [3]Genetic and Molecular Epidemiology Group, Spanish National Cancer Research Centre (CNIO), Madrid, Spain. [4]Instituto de Estudios de las Adicciones IEA-CEU, Universidad San Pablo-CEU, CEU Universities, Madrid, Spain. [5]Department of Allergy, Hospital Infantil Niño Jesús, Fib-HNJ, Madrid, Spain. [6]Instituto de Investigación Sanitaria-La Princesa, Madrid, Spain. [7]Pedriatic Allergy Unit, Allergy Service, Hospital General Universitario Gregorio Marañón, and Gregorio Marañón Health Research Institute, Madrid, Spain. [8]Department of Chemistry and Chemical Biology, McMaster University, Hamilton, ON, Canada. [9]Fundació per al Foment de la Investigació Sanitària i Biomèdica de la Comunitat Valenciana (FISABIO), Valencia, Spain. [10]CIBER en Epidemiología y Salud Pública, Madrid, Spain. [11]Joint Research Unit in Genomics and Health, Fundació per al Foment de la Investigació Sanitària i Biomèdica de la Comunitat Valenciana (FISABIO) and Institut de Biologia Integrativa de Sistemes (Universitat de València / Consejo Superior de Investigaciones Científicas), València, Spain. [12]These authors contributed equally: Tomás Clive Barker-Tejeda, Elisa Zubeldia-Varela, Andrea Macías-Camero. ✉e-mail: marina.perezgordo@ceu.es; alma.villasenor@ceu.es

