## [Peer Review File · Nature Communications]

REVIEWER COMMENTS

Reviewer #1. Infant-Mother microbiome / Metagenomics. (Remarks to the Author):

This manuscript provides an multi-omics gut microbiome analysis of a cohort of infants-mother-grandmother triads. The conclusion is that the gut microbiome and associated metabolome of the infants is very different from those of the (grand)mothers. Due to the mostly descriptive nature of the manuscript, the reader is not really getting any novel insights into a previously described phenomenon, and although the discussion tries to correlate some of the findings to ageing, this can not be taken as convincing evidence.

Further comments:

- Line 64-66: what exactly is meant with 'pre-birth gestation' impacts colonisation?
- Lines 78-81; the authors say that 'very few studies etc', but don't actually provide any references.
- Line 94-96: no distinct evidence is provided that their method works better than others, just that their method works. Perhaps the authors should make a statement regarding its relative performance.
- Line 111: Were the grandmothers living with the rest of the family, or how was this difference in contact time between families controlled for?
- Unclear that the exact same number of samples was run/analysed by 16S and Shotgun. Assuming it was? What I don't get, is having done shotgun, why the authors don't use this data for the taxonomic analysis. Why use amplicon for taxonomy? The fact they have the shotgun reads, it would be really nice to see if they get a different result for the generational differences depending on which dataset they use. Certain taxa may be represented differently by shotgun or 16S. (a slight personal bug is that they are stuck up at phyla level when they could get more important resolution at species and perhaps even strain level, in particular because shotgun data are available)
- In general for the analysis, its not fully clear whether they pooled all the samples from a given generation, or whether it was paired into triads. Seen as the number for each generation is different I assume the former. This could be important. If for example 6 of the infants were not related to any of the older generations then you are not really doing a relationship analysis. The analysis should probably be done on a paired basis rather than pooled.
- Lines 239-248: The authors present correlations between levels of measured metabolites and phyla that putatively produce them. But they have all the functional metagenomics, why don't they actually check for functional capacity rather than assume at such a broad taxonomic level?

- Line 234-237: The authors state that grandmothers are enriched for Deltaproteobacteria etc.. Is this relative to the infants, or enriched as a group itself? Because this slightly contradicts the point in line 229-230 that there were no significant differences between mothers and grandmothers.
- Line 681 states that 121 samples were sequenced, but line 207-208 gives a number of 128 (as does the figure6A)
- The authors state the number of reads obtained for 16S analysis, but how many reads (and mean) did they get for shotgun?
- Line 284: is this the correct figure reference?
- There wouldn't be a need for the correlation in Fig 9C ASV versus KO if they just used the shotgun taxonomy data.
- Figure9D: the authors speak about the outside line representing "expression levels" of a variable, not sure this is the correct terminology, perhaps measurement level?
- Line 297-298: Bifs are negatively correlated to most Kos. Does this mean they are not picking up/annotating functional genes and pathways related to Bifs in their shotgun data? Seen as Bifs are dominant in the infant, then that would be a problem for the KO analysis in the infant?
- Line 300-302, careful with the use of the word "proved"..
- Line 310-311: do we have this figure of the Cytoscape output?

Reviewer #2. Multi-omics data analyses. (Remarks to the Author):

Barker-Tejeda et al. used a combined metagenomics and metabolomics approach to characterize the microbiota and metabolic profiles in infants, their mothers and grandmothers. Univariate and multivariate analyses were performed to discriminate between infants, mothers and grandmothers using both metabolomics and metagenomics data. Infants were found to be significantly different than adults (mothers and grandmothers). Data integration also provided similar results with infants displaying different metabolite and bacterial patterns compared to adults. This work constitutes a significant amount of work in recruiting participants, collected sampling and generating the multiomics data. Additional analyses can be performed to gain additional insights from the data and answering research questions related to allergies and mode of delivery. The following comments should help improve this body of work.

Major comments

1. Previous studies have looked at changes in microbiota and metabolomics in relation to age (e.g. *Nutrients* 2020 12(12):3759. *Science advances* 2022, 8, 42). It is unclear how the inclusion of infants, mothers and grandmothers is advantageous compared to existing studies. The pairing between infant and mothers is not taken into account in any of the models making it difficult to interpret any infant/mother-microbiota/metabolite associations.
2. Although pairs of trios (infants, mothers and grandmothers) are used, this relationship is not accounted for in any of the modelling. For example, a linear mixed effects models can be used for univariate analysis in which the fixed effect can be the age-groups and the random effect can be specified by the trio.
3. Additive linear models of trios would be useful to identify patterns in which the microbiota and metabolites increase/decrease or stay the same across age groups.
4. Given that these infants are either breastfeeding or on formula it would be useful to compare the metabolite/microbiota between infants and mothers in relation to the mode of feeding. This may provide evidence between the infant/mother transmission of microbiota/metabolites if it exists. If not, a power-calculation may be used to determine the sample size required to show a given effect-size for a particular comparison.
5. Linking infants, mothers and grandmothers can also be accounted for in mixOmics by assuming this trio as repeated measurements. The multilevel argument can be used for this purpose prior to applying the multivariate models.
6. Adjustment for multiple correction should be reported in the same manner: the benjamini-hochberg correction is referred to as FDR, pFDR or adjusted q-value which can be confusing to the readers. Please use only one abbreviation such as FDR.

Minor comments

- 1) line 754 is a section about DIABLO, but lines 765-lines 771 provide code that is not related to the mixOmics library (please remove)

Reviewer #3. Metabolomics and Mass Spectrometry / Microbiome (Remarks to the Author):

The manuscript by Barker-Tejeda and colleagues describes a thorough examination of the gut microbiome in terms of 16S sequencing, shotgun sequencing and the metabolome. The cohort they examine is novel, looking at three generations. They also use bioinformatics to fuse these different datasets together which I think will be of interest to a number of groups trying to perform similar analyses - this is a part of the project that is very well performed. The authors are also to be

commended that they have made their data available through international repositories - both the genomics and metabolomics. All these aspects of the study are well performed and well described in the manuscript.

My major concern is that the main result is probably the obvious one. The infants show a different microbiome both in terms of sequencing and the metabolome compared with the mothers and grandmother groups. This must largely be down to diet given their stage in life, particularly given that at the first 12 months in life dietary consumption of milk will be very high and fibre understandably low. I would be really interested to see what happens with the infant group as they age and predict that they will look more like their mothers in the near future. This makes me think that its the switch in diets as the infants are weaned that might be most informative of the changes in the gut microbiome. Could this be investigated in this cohort? I think without this the results of the paper perhaps lack novel insight despite being well performed.

Specific comments:

Line 141. The MSI-CE-MS triple quadrupole method is an interesting and high throughput method. Could the authors include somewhere the MRMs used for the 14 metabolites.

Line 387: "In accordance, our study proved that BCAA conversion to BCFA is altered with age..." At this stage the authors have only found an association and not a proof unless they have other data/experiments to explain this association. They should re-write accordingly.

Line 412: "...and KOs such as those of the energetic metabolism..." As energetic metabolism is not a standard phrase I think it would be better to spell out what pathways they are referring to.

Line 430. "...as it is closely linked to the circumstances of each age period, making it difficult to dissociate the effects of diet and aging." I completely agree - this does dilute down the novelty of the paper though.

Line 481. The methods section is quite long in respect to the rest of the manuscript. I wonder if the section on GC-TOF MS could be shortened given its a published methods by the authors?

Line 531. Can the authors really perform batch correction given they run the infants, 'mothers' and 'grandmothers' samples all in individual batches? Some data needs to be presented as to how the batch correction works, otherwise we could be left with the conclusion that the infants are different because they were run in a different batch.

Line 552. I like the double IS method for the GC-TOF-MS and this is well described so others can try this.

Line 593: Is it summarised anywhere how many features remain after each step of the processing? This would be useful as a way to understand these QC/QA steps.

Line 737. Abbreviation of KO. Was this defined earlier? It was certainly used earlier.

REVIEWER COMMENTS

Reviewer #1. Infant-Mother microbiome / Metagenomics:

This manuscript provides an multi-omics gut microbiome analysis of a cohort of infants-mother-grandmother triads. The conclusion is that the gut microbiome and associated metabolome of the infants is very different from those of the (grand)mothers. Due to the mostly descriptive nature of the manuscript, the reader is not really getting any novel insights into a previously described phenomenon, and although the discussion tries to correlate some of the findings to ageing, this can not be taken as convincing evidence.

*We thank the reviewer for the feedback. We understand their perspective; however, in the revised version of the manuscript, we have emphasized the novelty of our research. Our study is unique as it applies a multi-omics approach (including **up to 4 omics analyses**) to a large number of fecal samples.*

We would like to stress the novelty of our approach for GC-QTOF-MS analysis, since we demonstrate here a successful batch analysis and quality control strategy, which might serve as a reference for many future studies with large cohorts. To our knowledge, this is the first time that such a methodology has been developed for large-scale GC-MS analysis.

We have used two metabolomics techniques and two metagenomics platforms to study the microbiota in a distinct design that includes three generations, allowing us to describe age-related changes. We believe that the value of this number of human infant fecal samples cannot be understated, and even more so when adding two generations of their maternal line. To the best of our knowledge, our study is the first to do this, as well as the first of this magnitude in Spain dedicated to exploring the associations between bacteria and metabolites in a population with the same geographical location, and similar cultural practices.

In addition, to the best of our knowledge, our manuscript is the first to correlate the bacterial taxa (phyla, genera, and species), the metabolites, and the functions (KOs) and to provide such an extensive multi-omic integration. We believe we have been capable of bringing together many aspects that were previously known, but that are presented here for the first time in the same samples and in such a comprehensive manner. This has been improved even further thanks to the comments from the reviewers by adding the shotgun taxonomy, the correlation between 16S rRNA and shotgun sequencing, and the new statistics linking the families. The extensive dataset we have generated – which has been expanded thanks to the reviewer's suggestions – provides a valuable resource for further exploration by researchers.

We acknowledge that the initial version did not adequately highlight our discoveries. However, with the revisions and new additions, we are confident that the novelty and significance of our research are now evident and compelling.

Further comments

1. Line 64-66: what exactly is meant with 'pre-birth gestation' impacts colonisation?

Thanks to the reviewer for raising this comment. Our meaning regarding this sentence was the idea that has been discussed in the scientific community that a microbiome is present in the fetus during pregnancy and that this may be a factor influencing colonization.

However, in a very recent publication by many of the top scientists in the microbiome field, it was suggested that what has been detected in newborns as prenatal period microbiota is likely contamination, as there should be no microbiome in a healthy fetus (Kennedy, K. M. et al. *Questioning the fetal microbiome illustrates pitfalls of low-biomass microbial studies*. *Nature* 613, 639–649 (2023).). Therefore, we have corrected this term and have added these new findings (lines 65 - 68).

2. Lines 78-81; the authors say that ‘very few studies etc’, but don’t actually provide any references.

We thank the reviewer for this comment. To the best of our knowledge, there are no studies that cover all the topics that are addressed in the present work: (i) study of the maternal lineage (including also the grandmothers) or any family lineage in a large-scale cohort; (ii) characterization of the fecal microbiome and metabolome at three time-points in life; and (iii) using a multi-omics approach.

Additionally, we would like to highlight that two online platforms have been developed in recent years to collect information on metabolites that have been shown to change with age. However, they are all characterized in either plasma/serum or urine, which again emphasized the importance and strength of our study (e.g., linking the fecal metabolome to aging).

1. HMDB Metabolome Database: Browsing Age Metabolomic Data (https://hmdb.ca/age_metabolomics)
Reference: Wishart DS, Feunang YD, Marcu A, et al. HMDB 4.0: the human metabolome database for 2018. *Nucleic Acids Res.* 2018;46(D1):D608-D617. doi:10.1093/nar/gkx1089

2. MetaboAge (<https://www.metaboage.info/>). Reference: Bucaciuc Mracica T, Anghel A, Ion CF, Moraru CV, Tacutu R, Lazar GA. MetaboAge DB: a repository of known ageing-related changes in the human metabolome. *Biogerontology*. 2020;21(6):763-771. doi:10.1007/s10522-020-09892-w

In agreement with the reviewer, and focusing on the novelty of this paper, we have made an extensive search and herein are listed various original articles that meet the criteria for multi-omics (metabolomics and metagenomics, either applying 16S rRNA gene or shotgun sequencing) studies in feces; comparing at least one of these two conditions: elder vs young population comparison or mothers vs infants cohorts – but not covering the three generations. We have added these references in the text, with a commentary highlighting the novelty of our paper (lines 83-84):

1. Djukovic, A., et al. (2022). *Lactobacillus* supports Clostridiales to restrict gut colonization by multidrug-resistant Enterobacteriaceae. *Nature communications*, 13(1), 5617. <https://doi.org/10.1038/s41467-022-33313-w>.

2. Poyet, M., et al. (2019). A library of human gut bacterial isolates paired with longitudinal multiomics data enables mechanistic microbiome research. *Nature medicine*, 25(9), 1442–1452. <https://doi.org/10.1038/s41591-019-0559-3>.

3. An, R., et al. (2019). Sugar Beet Pectin Supplementation Did Not Alter Profiles of Fecal Microbiota and Exhaled Breath in Healthy Young Adults and Healthy Elderly. *Nutrients*, 11(9), 2193. <https://doi.org/10.3390/nu11092193>.

4. Tuikhar, N., et al. (2019). Comparative analysis of the gut microbiota in centenarians and young adults shows a common signature across genotypically non-related populations. *Mechanisms of ageing and development*, 179, 23–35. <https://doi.org/10.1016/j.mad.2019.02.001>.

5. Brink, L. R., et al. (2020). Neonatal diet alters fecal microbiota and metabolome profiles at different ages in infants fed breast milk or formula. *The American journal of clinical nutrition*, 111(6), 1190–1202. <https://doi.org/10.1093/ajcn/nqaa076>.

6. Conta, G., et al. (2021). Longitudinal Multi-Omics Study of a Mother-Infant Dyad from Breastfeeding to Weaning: An Individualized Approach to Understand the Interactions Among Diet, Fecal Metabolome and Microbiota Composition. *Frontiers in molecular biosciences*, 8, 688440. <https://doi.org/10.3389/fmolb.2021.688440>.

7. Vatanen, T., et al. (2022). Mobile genetic elements from the maternal microbiome shape infant gut microbial assembly and metabolism. *Cell*, 185(26), 4921–4936.e15. <https://doi.org/10.1016/j.cell.2022.11.023>.

8. Boulangé, C. L., et al. (2023). An Extensively Hydrolyzed Formula Supplemented with Two Human Milk Oligosaccharides Modifies the Fecal Microbiome and Metabolome in Infants with Cow's Milk Protein Allergy. *International journal of molecular sciences*, 24(14), 11422. <https://doi.org/10.3390/ijms241411422>.

3. Line 94-96: no distinct evidence is provided that their method works better than others, just that their method works. Perhaps the authors should make a statement regarding its relative performance.

We thank the reviewer for raising this comment. Our laboratory has extensive experience regarding the traditional derivatization of biological samples for their analysis in GC-MS (DOI: https://doi.org/10.1007/978-1-61737-985-7_11). From 2010 to date, we have been applying the traditional methods and we know that they are difficult to implement in large-scale analyses. Therefore, one of our aims was to develop a new strategy that could overcome this problem. The details regarding its performance are included in M&M section in lines 501-509.

“Prior to this study, we adapted a derivatization method that allows for shorter derivatization times <15 min¹⁹ compared to the 16 hours of traditional approaches^{20,23}. It also enhances the coverage of key metabolites of the host-gut microbiota co-metabolism, such as SCFA, which have poor recoveries in the traditional derivatization reactions since they require anhydrous media and samples must be evaporated up to total dryness. We applied this methodology to a QTOF mass analyzer, a high-resolution equipment well-known for its powerful capabilities for unveiling unknown compounds. It allows for the determination of accurate mass in all detected mass-to-charge (m/z) fragments, which enormously enhances the selectivity and the sensitivity of the analysis and leads to a much more reliable identification^{20,74}.”

4. Line 111: Were the grandmothers living with the rest of the family, or how was this difference in contact time between families controlled for?

It was not controlled for; but based on our knowledge of the area where the samples were recruited, cohabitation between multiple generations is uncommon. This was confirmed by the clinicians that participated in this study, but it is an interesting variable to consider in future studies.

5. Unclear that the exact same number of samples was run/analysed by 16S and Shotgun. Assuming it was? What I don't get, is having done shotgun, why the authors don't use this data for the taxonomic analysis. Why use amplicon for taxonomy? The fact they have the shotgun reads, it would be really nice to see if they get a different result for the generational differences depending on which dataset they use. Certain taxa may be represented differently by shotgun or 16S. (a slight personal bug is that they are stuck up at phyla level when they could get more important resolution at species and perhaps even strain level, in particular because shotgun data are available)

We thank the reviewer for pointing this out. To clarify, the same samples were indeed analyzed by both 16S rRNA gene and shotgun sequencing. The reason behind using 16S rRNA gene sequencing for the taxonomic analysis is that our first goal was to integrate three types of omics technologies (metabolomics, 16S rRNA and shotgun) to incorporate and correlate three different types of data from three different platforms. If only shotgun metagenomics had been used, the taxonomic and functional data would have come from the same analysis and thus would have correlated better than the independent analyses performed in this study.

However, we agree with the reviewer that shotgun sequencing has more taxonomic resolution. Thus, following their suggestion, taxonomic analysis using shotgun has been carried out for phyla, genera, and species. These results were incorporated in the manuscript and have been compared to the 16S rRNA analysis (Figures 7 and S7, and Tables S5, S6 and S7). We observed similar results for phyla and genera with both genomic techniques, in agreement with other studies comparing them both (DOI: <https://doi.org/10.1038/s41598-021-82726-y> and <https://doi.org/10.1038/s41598-022-07995-7>).

Consequently, for the mixOmics analysis, we kept the taxonomic information obtained by the 16S rRNA over shotgun sequencing, as we observed (i) a high correlation for the 85 most abundant matched genera (with a minimum abundance of at least 1000 counts among samples) showed by 90% of them being significant and all positively correlated ($p > 0.70$ in 70% of the genera, Table S14); and (ii) to include information from three independent datasets.

In addition, following the reviewer's comments, we must clarify that the 16S rRNA data were also obtained for genera, not just phyla, and it is with genera that most of our results are discussed. However, to dig up more in the shotgun sequencing data, the statical analysis by species was carried out and has been included in the main manuscript to provide more information (Figure S7, Table S8). Finally, new correlations between metabolites (GABA and SCFA) with species have been added to the manuscript (Figure S8 and Tables S9 and S10). We hope that the new version is now clear and complete.

6. In general for the analysis, its not fully clear whether they pooled all the samples from a given generation, or whether it was paired into triads. Seen as the number for each generation is different I assume the former. This could be important. If for example 6 of the infants were not related to any of the older generations then you are not really doing a relationship analysis. The analysis should probably be done on a paired basis rather than pooled.

We thank the reviewer for raising this comment. Indeed, in the first analysis the samples were pooled into the three different groups, but thanks to their comment and other reviewers we have changed the univariate statistics to address the dependence between samples (Infants, Mothers, and Grandmothers), and the situation of relatives. As we have a few situations where two infants have the same mother and grandmother (i.e. a couple families with twins), we have applied a linear mixed-effects model (GLM) using the MaAsLin2 script, where the generation was specified as the fixed effect and the family as the random effect. In addition, the False Discovery Rate (FDR) was performed, and statistical significance was set at 95% level ($FDR <$

0.05). We applied this test for the metabolomics data (GC-QTOF-MS and MSI-CE-MS), the taxonomic data from metagenomics (16S rRNA gene and shotgun sequencing) – i.e. phyla, genera, and species –, and the functional data from shotgun sequencing (KEGG and BRITE B and C functional hierarchies). We have added this information to the manuscript, and you can find it in lines 699-705.

We have observed that most of the changes continue to be significant, and that additional information proved significant as well. We consider that these changes reinforce our original conclusions and make the manuscript more straightforward.

7. Lines 239-248: The authors present correlations between levels of measured metabolites and phyla that putatively produce them. But they have all the functional metagenomics, why don't they actually check for functional capacity rather than assume at such a broad taxonomic level?

As mentioned in our previous comment (comment 5), we need to clarify that we do not only correlate with phyla, but mainly with genera. However, following the reviewer's suggestions, new correlations between metabolites (GABA and SCFA) and species have been added to the manuscript (Figure S8 and Tables S9 and S10). We hope that the new version gives more specific information regarding the relationship between the species and the metabolites.

8. Line 234-237: The authors state that grandmothers are enriched for Deltaproteobacteria etc.. Is this relative to the infants, or enriched as a group itself? Because this slightly contradicts the point in line 229-230 that there were no significant differences between mothers and grandmothers.

We thank the reviewer for this comment. The apparent contradiction is due to the results coming from two different analyses: the first (**lines 229-230 of the original manuscript**) was from ANCOM-BC, which is stricter and corrects by FDR, so no genera are significant between mothers and grandmothers. This result is confirmed by the new GLM analysis we have applied. The other one (**lines 234-237 of the original manuscript**) is based on LEfSe, which looks for enrichment in a different way (in each group compared to the rest) and shows indeed that the grandmothers are enriched in these taxa as a group themselves (compared to both mothers and infants).

We have edited the text to make it clearer (lines 242-245):

“These results confirm that the main genera **enriched** in the Infant microbiota – compared to the other two groups – are *Bifidobacterium* and *Lactobacillus*, while Mothers are enriched in the Clostridia class (*Clostridium* III group and *Blautia* genus), and Grandmothers have higher levels of the Deltaproteobacteria class and *Verrumicrobia* phylum.”

9. Line 681 states that 121 samples were sequenced, but line 207-208 gives a number of 128 (as does the figure6A)

We thank the reviewer for pointing this out. These numbers are correct and are better explained in Figure S1A, S2 and Table S15. All 128 samples from the subset were sequenced, and the final dataset included 121 after removal of age and statistical outliers. It is true that the methods section was confusing, so we have revised it. We hope that it is clearer now.

10. The authors state the number of reads obtained for 16S analysis, but how many reads (and mean) did they get for shotgun?

Following the reviewer's suggestion, we have added "Likewise, for shotgun sequencing, a total of 3,569,152,984 paired-end reads with an average of 3,379,879 sequences per sample were obtained" (lines 215-217).

11. Line 284: is this the correct figure reference?

Yes, we meant to highlight the reasoning behind the number of samples that was used for the integration (i.e., the ones that were measured in all techniques and were not outliers). The figure caption from Figure S1B explains this number.

We have also cited Table S15, which specifies in the further right column which samples were included. We have added a row at the end with a "total n", so it is easier to follow. We have edited the text and hope it is now clearer (lines 301-302): "Thus, we used the data from the samples that were measured successfully in all the analyses (N=113, see Figure S1B and Table S15 for explanations)".

12. There wouldn't be a need for the correlation in Fig 9C ASV versus KO if they just used the shotgun taxonomy data.

We thank the reviewer for his comment. The three datasets are included in mixOmics because they provide different information: taxonomy (ASVs), functional information (KOs), and metabolites. In this way, we can increase the reliability of existing correlations and ensure that they are not biased by the methodology or technique used, as previously explained. However, to confirm that both datasets (16S rRNA gene and shotgun sequencing) provide the same taxonomic information, we performed a correlation analysis using Pearson's correlation, where we observed that almost 90% were significantly and positively correlated. We have included this analysis in lines 297-299 and Table S14.

13. Figure 9D: the authors speak about the outside line representing "expression levels" of a variable, not sure this is the correct terminology, perhaps measurement level?

The reviewer is correct. The terminology has been changed to "measured levels" in line 1086.

14. Line 297-298: Bifs are negatively correlated to most Kos. Does this mean they are not picking up/annotating functional genes and pathways related to Bifs in their shotgun data? Seen as Bifs are dominant in the infant, then that would be a problem for the KO analysis in the infant?

Thank you for the comment. When we say "most of the KOs" in the discussion, we mean from the top ones selected by mixOmics. Also, the fact that they are negatively correlated means that while Bifidobacterium levels are very high in infants, these specific KOs are much lower, while the opposite is true in adults. However, this does not mean that Bifidobacterium pathways are missing. To clarify this, the text has been changed to "negatively correlated with most KOs selected by the mixOmics analysis from this subset" in line 316. We hope that the text and their meaning is clear now.

15. Line 300-302, careful with the use of the word “proved”

The reviewer is right, this is not the correct word for this context. We have changed it to “classified the Infants almost perfectly into their own cluster” (lines 318-320).

16. Line 310-311: do we have this figure of the Cytoscape output?

This figure should appear as part of the Supplementary Data (“Figure_S_9_Cytoscape”), which you may need to download separately. We hope that you can easily find it in the updated version.

Reviewer #2. Multi-omics data analyses:

Barker-Tejeda et al. used a combined metagenomics and metabolomics approach to characterize the microbiota and metabolic profiles in infants, their mothers and grandmothers. Univariate and multivariate analyses were performed to discriminate between infants, mothers and grandmothers using both metabolomics and metagenomics data. Infants were found to be significantly different than adults (mothers and grandmothers). Data integration also provided similar results with infants displaying different metabolite and bacterial patterns compared to adults. **This work constitutes a significant amount of work in recruiting participants, collected sampling and generating the multiomics data. Additional analyses can be performed to gain additional insights from the data and answering research questions related to allergies and mode of delivery.** The following comments should help improve this body of work.

Thank you for the kind words and for the nice suggestions to improve our work, they are greatly appreciated. We hope that the changes that we have made have improved substantially the manuscript. We have integrated the metagenomics and metabolomics data to gain a comprehensive understanding of the relationships between microbiota composition and metabolic profiles. This data integration provided consistent results, confirming the distinct patterns observed in infants compared to adults. We have expanded upon this integration in our revised manuscript to emphasize the significance of these findings.

Major comments

1. Previous studies have looked at changes in microbiota and metabolomics in relation to age (e.g. *Nutrients* 2020 12(12):3759. *Science advances* 2022, 8, 42). It is unclear how the inclusion of infants, mothers and grandmothers is advantageous compared to existing studies. The pairing between infant and mothers is not taken into account in any of the models making it difficult to interpret any infant/mother-microbiota/metabolite associations.

We thank the reviewer for raising this comment. We have reviewed and included these references about these kinds of studies in the main manuscript. However, to the best of our knowledge, there are no studies that cover all the topics that are addressed in the present work: (i) study of the maternal lineage (infants, their mothers, and their grandmothers) in a large-scale cohort; (ii) characterization of the fecal microbiome and metabolome at three time-points in life; and (iii) using a multi-omics approach.

The inclusion of the three generations of families has the advantage of reducing the genetic variability of the study population. It is true that without the correct statistical analysis the family relations are not leveraged to their full potential. It is important for statistical analysis to address the non-independence between samples belonging to the same group and to integrate situations such as two infants with the same mother and grandmother (i.e. a couple families with twins).

To this effect, we have applied the MaAsLin2 script to run the GLM, where the generation was specified as the fixed effect and the family as the random effect. In addition, the False Discovery Rate (FDR) was performed, and statistical significance was set at 95% level ($FDR < 0.05$). We applied this test for the metabolomics data (GC-QTOF-MS and MSI-CE-MS), the taxonomic data from metagenomics (16S rRNA gene and shotgun sequencing) – i.e. phyla, genera, and species –, and the functional data from shotgun sequencing (KEGG and BRITE B and C functional hierarchies) (lines 699-705).

Following the reviewer's comments, we have strengthened the novelty of this manuscript, and we have changed the results of the statistical analysis. Notably, the majority of the significant values expressed in the first version of the manuscript continue to be relevant, confirming our findings and strengthening our biological interpretation. In addition, we have included the analysis of species by shotgun sequencing in the manuscript. We hope that after all these changes, we have clarified the message.

2. Although pairs of trios (infants, mothers and grandmothers) are used, this relationship is not accounted for in any of the modelling. For example, a **linear mixed effects models** can be used for univariate analysis in which the fixed effect can be the age-groups and the random effect can be specified by the trio.

We thank the reviewer for this suggestion; as said in the previous comment, we have changed the statistical approach following their recommendations.

3. **Additive linear models of trios** would be useful to identify patterns in which the microbiota and metabolites increase/decrease or stay the same across age groups.

Following the reviewer's advice, we have tried to compare by trios using a repeated-measures ANOVA test (for the families presenting data of the 3 individuals). We started performing this test with the metabolomic data (for both GC-QTOF-MS and MSI-CE-MS data). We have found that the information was very similar: 92% of the metabolites continued to be significant after this analysis, including all the main ones that constitute our biological interpretation. However, considering the non-independence between samples belonging to the same group and in order to integrate situations such as two infants with the same mother and grandmother (i.e. a couple families with twins), we have decided that it is more accurate to work using a mixed effect model instead, with the generation as the fixed effect and the family as the random effect, as explained above. We hope this has cleared the statistical question.

4. Given that these infants are either breastfeeding or on formula **it would be useful to compare the metabolite/microbiota between infants and mothers in relation to the mode of feeding.** This may provide evidence between the infant/mother transmission of microbiota/metabolites if it exists. If not, a power-calculation may be used to determine the sample size required to show a given effect-size for a particular comparison.

We thank the reviewer for this suggestion. We have tried to do this analysis applying one-way repeated measures tests, but the results were very complicated to interpret. In addition, we believe that the experimental design of this study was not meant to prove this hypothesis. Moreover, the concept of a "metabolite transmission" is very complex since metabolites are more host dependent and cannot be "transmitted" as such (they come from diet, individual's metabolism, etc).

Additionally, it is difficult to draw conclusions from the diet variable since we can only know what diet the infant was taking at the moment of sample collection. Almost all of the infants probably had breastfeeding at some early stage, as this is quite common in Spain. So, we cannot know for how long the baby was breastfeeding, how much time has passed since they stopped, etc. All these questions are very interesting to design future experiments tailored to further investigate the relationship between infant feeding and the interaction between their metabolism and their microbiota.

5. Linking infants, mothers and grandmothers can also be accounted for in mixOmics by assuming this trio as repeated measurements. The multilevel argument can be used for this purpose prior to applying the multivariate models.

We thank the reviewer for this suggestion; we have tried to do it but unfortunately, we were unable to carry out this analysis. We hope that with the additional analyses we have performed our results are clear and convincing.

6. Adjustment for multiple correction should be reported in the same manner: the benjamini-hochberg correction is referred to as FDR, pFDR or adjusted q-value which can be confusing to the readers. Please use only one abbreviation such as FDR.

We agree with the reviewer on this point. Therefore, we have changed everything to FDR to avoid any confusion for the readers.

Minor comments

1. line 754 is a section about DIABLO, but lines 765-lines 771 provide code that is not related to the mixOmics library (please remove)

Thanks to the reviewer for this annotation, we have removed the code.

Reviewer #3. Metabolomics and Mass Spectrometry / Microbiome:

The manuscript by Barker-Tejeda and colleagues describes a thorough examination of the gut microbiome in terms of 16S sequencing, shotgun sequencing and the metabolome. The cohort they examine is novel, looking at three generations. They also use bioinformatics to fuse these different datasets together which I think will be of interest to a number of groups trying to perform similar analyses - this is a part of the project that is very well performed. The authors are also to be commended that they have made their data available through international repositories - both the genomics and metabolomics. **All these aspects of the study are well performed and well described in the manuscript.**

My major concern is that **the main result is probably the obvious one.** The infants show a different microbiome both in terms of sequencing and the metabolome compared with the mothers and grandmother groups. **This must largely be down to diet** given their stage in life, particularly given that at the first 12 months in life dietary consumption of milk will be very high and fibre understandably low. **I would be really interested to see what happens with the infant group as they age and predict that they will look more like their mothers in the near future.** This makes me think that its the switch in diets as the infants are weaned that might be most informative of the changes in the gut microbiome. **Could this be investigated in this cohort?** I think without this the results of the paper perhaps lack novel insight despite being well performed.

We thank this reviewer for their kind comments and appreciation of the positive aspects of our work. We understand the point of view of the reviewer regarding the prediction in time; however, to answer the question raised, we would have to follow these infants for more than 20 years from now, which is not feasible. This is one of the limitations of the study stated in the discussion part.

We would like to stress the novelty of our approach for GC-QTOF-MS analysis, since we demonstrate here a successful batch analysis and quality control strategy, which might serve as a reference for many future studies with large cohorts. To our knowledge, this is the first time that such a methodology has been developed for large-scale GC-MS analysis.

We believe that the value of this number of human infant fecal samples cannot be understated, and even more so when adding two generations of their maternal line. To the best of our knowledge, our study is the first to do this, as well as the first of this magnitude in Spain dedicated to exploring the associations between bacteria and metabolites in a population with the same geographical location, and similar cultural practices.

In addition, to the best of our knowledge, our manuscript is the first to correlate the bacterial taxa (phyla, genera, and species), the metabolites, and the functions (KOs) and to provide such an extensive multi-omic integration. We believe we have been capable of bringing together many aspects that were previously known, but that are presented here for the first time in the same samples and in such a comprehensive manner. This has been improved even further thanks to the comments from the reviewers by adding the shotgun taxonomy, the correlation between 16S rRNA and shotgun sequencing, and the new statistics linking the families. The extensive dataset we have generated – which has been expanded thanks to the reviewer's suggestions – provides a valuable resource for further exploration by researchers, as the reviewer kindly points out.

We acknowledge that the initial version did not adequately highlight our discoveries. However, with the revisions and new additions, we are confident that the novelty and significance of our research are now evident and compelling.

Specific comments

1. Line 141. The MSI-CE-MS triple quadrupole method is an interesting and high throughput method. Could the authors include somewhere the MRMs used for the 14 metabolites.

We thank the reviewer for their suggestion, but there is a confusion. In our work, MSI-CE-MS was applied for targeted search of a panel of polar metabolites from stool extracts when using time-of-flight mass spectrometry with full-scan acquisition under positive or negative ion mode as detailed elsewhere (Shanmuganathan et al. Nature Methods 2021, 16:1966). As a result, we did not use multiple reaction monitoring (MRM)-based data acquisition with a triple quadrupole mass spectrometer. All fecal metabolites measured by MSI-CE-MS were annotated based on their accurate mass, relative migration time and ionization mode (m/z:RMT:p/n) and confirmed by spiking with authentic standards (i.e., co-migration) to enable their reliable identification. The exact masses are already provided in Table S3.

2. Line 387: "In accordance, our study proved that BCAA conversion to BCFA is altered with age..." At this stage the authors have only found an association and not a proof unless they have other data/experiments to explain this association. They should re-write accordingly.

We agree with the reviewer. We have changed "proved" to "suggests" in line 404.

3. Line 412: "...and KOs such as those of the energetic metabolism..." As energetic metabolism is not a standard phrase I think it would be better to spell out what pathways they are referring to.

We thank the reviewer for raising this issue. Indeed, we used energetic metabolism since it is the BRITE term provided by KEGG Database (this has been now clarified in the text). Additionally, we have also rewritten this section and added specific examples for further clarification (lines 430-436). We hope it is clearer now.

4. Line 430. "...as it is closely linked to the circumstances of each age period, making it difficult to dissociate the effects of diet and aging." I completely agree - this does dilute down the novelty of the paper though.

We thank the reviewer for raising this comment. However, as we said in the manuscript, it is impossible to dissociate the effects of diet and age. This is a real-life situation (we cannot have adults with a milk-based diet or solid-based for infants).

Additionally, we would like to highlight that two online platforms have been developed in recent years to collect information on metabolites that have been shown to change with age. However, they are all characterized in either plasma/serum or urine, which again emphasized the importance and strength of our study (e.g., linking the fecal metabolome to aging).

*1. HMDB Metabolome Database: Browsing Age Metabolomic Data (https://hmdb.ca/age_metabolomics)
Reference: Wishart DS, Feunang YD, Marcu A, et al. HMDB 4.0: the human metabolome database for 2018. Nucleic Acids Res. 2018;46(D1):D608-D617. doi:10.1093/nar/gkx1089*

2. *MetaboAge* (<https://www.metaboage.info/>). Reference: Bucaciuc Mracica T, Anghel A, Ion CF, Moraru CV, Tacutu R, Lazar GA. *MetaboAge DB: a repository of known ageing-related changes in the human metabolome*. *Biogerontology*. 2020;21(6):763-771. doi:10.1007/s10522-020-09892-w

In agreement with the reviewer, and focusing on the novelty of this paper, we have made an extensive search and herein are listed various original articles that meet the criteria for multi-omics (metabolomics and metagenomics, either applying 16S rRNA gene or shotgun sequencing) studies in feces; comparing at least one of these two conditions: elder vs young population comparison or mothers vs infants cohorts – but not covering the three generations (lines 83-84). We hope that the revised version of the manuscript will be clear on this point.

5. Line 481. The methods section is quite long in respect to the rest of the manuscript. I wonder if the section on GC-TOF MS could be shortened given its a published methods by the authors?

We agree with the reviewer, the GC-QTOF-MS section was quite long. We have reduced it and left only the most important parameters. We hope the final version is satisfactory.

6. Line 531. Can the authors really perform batch correction given they run the infants, 'mothers' and 'grandmothers' samples all in individual batches? Some data needs to be presented as to how the batch correction works, otherwise we could be left with the conclusion that the infants are different because they were run in a different batch.

We thank the reviewer for bringing this point up. We are confident in the results for several reasons:

- (i) *The QC samples, which are extensively discussed to be a mixture of samples from the three age groups (in the methods section and in Figure S10). These samples allow us to perform a normalization that results in the correction of the batch effect. This is proven when plotting the Total Useful Signal (TUS) and the two internal standards (IS), as can be seen in Figure S3B and D. In addition, after normalization the QC samples cluster almost perfectly in the PCA model, as can be seen in Figure 1C.*
- (ii) *In any case, the batch strategy was only applied in GC-QTOF-MS. Samples were randomized extensively in MSI-CE-TOF-MS so no batch effects could appear between the age groups, and in metagenomics they were analysed in only one batch. Specifically, some metabolites were measured in both GC-QTOF-MS and MSI-CE-TOF-MS, and the results from their comparison are presented in Figure S5, showing the correlations, and the boxplots for each technique. The results are very similar, and the slight differences could be due to methodology bias rather than the batch strategy. All in all, the fact that the differences are very similar in the other datasets makes sure we can trust the results from GC-QTOF-MS.*

We have also edited the text in several points to make this clearer: Line 124-125: “(...) using a common quality control sample (QC) made from pooled samples from the three age groups and analyzed during all batches (Figure 1A)”. We hope this revised version is satisfactory.

7. Line 552. I like the double IS method for the GC-TOF-MS and this is well described so others can try this.

We thank the reviewer for the nice praise of this methodology.

8. Line 593: Is it summarised anywhere how many features remain after each step of the processing?
This would be useful as a way to understand these QC/QA steps.

Due to the extension of the main manuscript, we have included these outcomes of the QA steps in the materials and methods section (lines 587-598). We hope this information makes the process clearer.

9. Line 737. Abbreviation of KO. Was this defined earlier? It was certainly used earlier.

Thanks for this comment, indeed KO was defined in line 271. We decided to define the abbreviations again in Materials and Methods for greater clarity. If needed, we will define each term only once if the editorial team requires it.

REVIEWERS' COMMENTS

Reviewer #1 (Remarks to the Author):

The authors have gone through a lot of effort to counter the reviewers' comments. In particular the manuscript now emphasizes the high-throughput approach and data integration efforts, rather than trying to focus on the actual gained insights.

I have some further comments listed below to improve the manuscript:

Lines 67-68; 'mechanisms' is a bit of an odd word here, perhaps delete 'are the main mechanisms that'; the authors may also include more recent publications on this topic, where other factors affecting mother-to-baby transmission have been identified, such as full/pre term birth and maternal membrane rupture (PMID: 37230981).

Lines 229-231; line 254; the phylum nomenclature has recently been updated.

Line 346-347; the statement 'focusing on the maternal transference to the infants' does not make sense as this was not assessed as far as I can see.

Line 381; GABA production among Bifidobacterium species is best studied for *B. adolescentis* strains, but I am unaware of *B. bifidum*, *B. breve* and *B. longum* producing this compound. The authors may check this and add relevant reference(s).

Lines 397-398; 'diet is based on milk (whether breastmilk, formula or a combination of both), which is mainly composed of complex oligosaccharides', is an incorrect statement as the main carbohydrate in milk is lactose (which is the second most abundant solid component of milk after protein).

Figure 4B (and accompanying text in the manuscript); the central image displays the TCA cycle and although this is indeed a very important energy generating pathways for many microorganisms that perform respiration, it may be less relevant to gut commensals which live in an environment with very little oxygen. In such cases, the TCA cycle is generally partially present and serves to generate precursors for biosynthetic purposes. This section of the results should therefore take this into consideration.

Reviewer #2 (Remarks to the Author):

The authors have addressed my comments and have improved this manuscript. I have no additional concerns.

REVIEWER COMMENTS

Reviewer #1:

The authors have gone through a lot of effort to counter the reviewers' comments. In particular the manuscript now emphasizes the high-throughput approach and data integration efforts, rather than trying to focus on the actual gained insights.

We would like to thank the reviewer very much for their kind words and the positive reaction to our study.

Further comments

1. Lines 67-68: 'mechanisms' is a bit of an odd word here, perhaps delete 'are the main mechanisms that'; the authors may also include more recent publications on this topic, where other factors affecting mother-to-baby transmission have been identified, such as full/pre term birth and maternal membrane rupture (PMID: 37230981).

Thanks to the reviewer for raising this comment. Following their suggestion, we have deleted that part of the manuscript and the text now reads: "The initial microbial seeding of the infant comes predominantly from the maternal side since vaginal delivery and breastfeeding influence the primary colonization of the gut microbiota, among other factors" (lines 65-66).

Additionally, we have added the following references regarding the influence of different maternal factors, other than the mode of delivery and breastfeeding, on the early development of the gut microbiota (line 66):

1. Jeong S. (2022). Factors influencing development of the infant microbiota: from prenatal period to early infancy. *Clinical and experimental pediatrics*, 65(9), 439–447. <https://doi.org/10.3345/cep.2021.00955>
2. Bogaert, D., van Beveren, G. J., de Koff, E. M., Lusarreta Parga, P., Balcazar Lopez, C. E., Koppensteiner, L., Clerc, M., Hasrat, R., Arp, K., Chu, M. L. J. N., de Groot, P. C. M., Sanders, E. A. M., van Houten, M. A., & de Steenhuijsen Pitors, W. A. A. (2023). Mother-to-infant microbiota transmission and infant microbiota development across multiple body sites. *Cell host & microbe*, 31(3), 447–460.e6. <https://doi.org/10.1016/j.chom.2023.01.018>.
3. Feehily, C., O'Neill, I. J., Walsh, C. J., Moore, R. L., Killeen, S. L., Geraghty, A. A., Lawton, E. M., Byrne, D., Sanchez-Gallardo, R., Nori, S. R. C., Nielsen, I. B., Wortmann, E., Matthews, E., O'Flaherty, R., Rudd, P. M., Groeger, D., Shanahan, F., Saldova, R., McAuliffe, F. M., Van Sinderen, D., ... Cotter, P. D. (2023). Detailed mapping of Bifidobacterium strain transmission from mother to infant via a dual culture-based and metagenomic approach. *Nature communications*, 14(1), 3015. <https://doi.org/10.1038/s41467-023-38694-0>.

We hope that the new version is now clear and complete.

2. Lines 229-231: line 254; the phylum nomenclature has recently been updated.

The reviewer is correct. However, because the previous informal names are well established and currently accepted, we have retained that nomenclature and added the one recently published by the International Code of Nomenclature of Prokaryotes (ICNP) (doi:10.1099/ijsem.0.005056). This has been done both in the manuscript (each time a new phylum was mentioned: lines 224-225, line 230, and lines 246-247) and in the tables (Table S05 and S06).

3. Line 346-347: the statement ‘focusing on the maternal transference to the infants’ does not make sense as this was not assessed as far as I can see.

We agree with the reviewer on this point. Therefore, this sentence has been removed from the text, resulting in (lines 333-335): “To our knowledge, this is the first study to characterize the fecal microbiome and metabolome of Infants, their Mothers, and Grandmothers, leaving out heterogeneity due to the genetic line”.

4. Line 381: GABA production among Bifidobacterium species is best studied for B. adolescentis strains, but I am unaware of B. bifidum, B. breve and B. longum producing this compound. The authors may check this and add relevant reference(s).

We thank the reviewer for raising this comment. Indeed, GABA production is extensively described in the literature for both Bifidobacterium adolescentis (doi:10.1038/s41598-020-70986-z) and B. dentium (doi:10.1111/nmo.12904) strains. However, during the biological interpretation of our combined metabolomics and metagenomics data (metabolites, taxonomy, and functionality), both published literature and omics databases (e.g. HMDB, KEGG) were used. Regarding the latter, the correlation between GABA and B. bifidum, B. breve or B. longum was described using the Virtual Metabolic Human database (doi:10.1093/nar/gky992) as can be seen from the following screenshot:

Organism	Phylum	Class
Bifidobacterium bifidum BGN4	Actinobacteria	Actinobacteria
Bifidobacterium bifidum NCIMB 41171	Actinobacteria	Actinobacteria
Bifidobacterium bifidum PBI 2010	Actinobacteria	Actinobacteria
Bifidobacterium bifidum S17	Actinobacteria	Actinobacteria
Bifidobacterium boum DSM 20432	Actinobacteria	Actinobacteria
Bifidobacterium breve HPH0326	Actinobacteria	Actinobacteria
Bifidobacterium breve UCC2003 (NCIMB8807)	Actinobacteria	Actinobacteria
Bifidobacterium coryneforme DSM 20216	Actinobacteria	Actinobacteria
Bifidobacterium dentium ATCC 27678	Actinobacteria	Actinobacteria
Bifidobacterium gallium DSM 20093	Actinobacteria	Actinobacteria
Bifidobacterium kashivaoense DSM 21854	Actinobacteria	Actinobacteria
Bifidobacterium longum DJO10A	Actinobacteria	Actinobacteria
Bifidobacterium longum E18	Actinobacteria	Actinobacteria
Bifidobacterium longum infantis 157F-NC	Actinobacteria	Actinobacteria
Bifidobacterium longum infantis ATCC 15697	Actinobacteria	Actinobacteria
Bifidobacterium longum longum ATCC 55613	Actinobacteria	Actinobacteria
Bifidobacterium longum longum BSMN68	Actinobacteria	Actinobacteria
Bifidobacterium longum longum CCLUG 52486	Actinobacteria	Actinobacteria
Bifidobacterium longum longum JCM 1217	Actinobacteria	Actinobacteria
Bifidobacterium longum longum JFM201	Actinobacteria	Actinobacteria
Bifidobacterium longum NCC2705	Actinobacteria	Actinobacteria

Following the reviewer's comment, we have added this reference to the publication describing the Virtual Metabolic Human database, along with the following text (lines 367-368): "... and species (Bifidobacterium bifidum, B. breve, B. longum) – the latter one according to the Virtual Metabolic Human database – ...". We hope that this additional information is now clear and convincing.

5. Lines 397-398; 'diet is based on milk (whether breastmilk, formula or a combination of both), which is mainly composed of complex oligosaccharides', is an incorrect statement as the main carbohydrate in milk is lactose (which is the second most abundant solid component of milk after protein).

Thanks to the reviewer for this annotation. Accordingly, the word "complex" has been deleted and the text has been changed to (382-384): "In the case of Infants, their diet is based on milk (whether breastmilk, formula or a combination of both), which is mainly composed of oligosaccharides that are metabolized by different strains of Bifidobacterium".

6. Figure 4B (and accompanying text in the manuscript); the central image displays the TCA cycle and although this is indeed a very important energy generating pathways for many microorganisms that perform respiration, it may be less relevant to gut commensals which live in an environment with very little oxygen. In such cases, the TCA cycle is generally partially present and serves to generate precursors for biosynthetic purposes. This section of the results should therefore take this into consideration.

We thank the reviewer for their comment, and we believe that they are right on this point. Since the metabolites displayed in Figure 4A do not only belong to the TCA cycle, but also to other energy production pathways even in an anaerobic context (e.g. glucose), we have modified the title of Figure 4A to "Energetic and biosynthetic pathways". Additionally, we have also addressed this consideration in the main text, both in the results (lines 192-195) and discussion (lines 355-357) sections.

We hope everything is clearer now.

Reviewer #2:

The authors have addressed my comments and have improved this manuscript. I have no additional concerns.

We would like to thank the reviewer very much for their kind words and the positive reaction to our study.